# Pancreatic α and β cells are globally phase-locked

Huixia Ren [1,2,6], Yanjun Li[1,3,6], Chengsheng Han[3,6], Yi Yu[1], Bowen Shi[2], Xiaohong Peng[3], Tianming Zhang[4], Shufang Wu[1], Xiaojing Yang[1,2], Sneppen Kim [5], Liangyi Chen [2,3✉] & Chao Tang [1,2✉]

The $Ca^{2+}$ modulated pulsatile glucagon and insulin secretions by pancreatic α and β cells play a crucial role in glucose homeostasis. However, how α and β cells coordinate to produce various $Ca^{2+}$ oscillation patterns is still elusive. Using a microfluidic device and transgenic mice, we recorded $Ca^{2+}$ signals from islet α and β cells, and observed heterogeneous $Ca^{2+}$ oscillation patterns intrinsic to each islet. After a brief period of glucose stimulation, α and β cells' oscillations were globally phase-locked. While the activation of α cells displayed a fixed time delay of ~20 s to that of β cells, β cells activated with a tunable period. Moreover, islet α cell number correlated with oscillation frequency. We built a mathematical model of islet $Ca^{2+}$ oscillation incorporating paracrine interactions, which quantitatively agreed with the experimental data. Our study highlights the importance of cell-cell interaction in generating stable but tunable islet oscillation patterns.

[1] Center for Quantitative Biology, Peking University, Beijing 100871, China. [2] Peking-Tsinghua Center for Life Sciences, Peking University, Beijing 100871, China. [3] Institute of Molecular Medicine, School of Future Technology, National Biomedical Imaging Center, Peking University, Beijing 100871, China. [4] Yuanpei College, Peking University, Beijing 100871, China. [5] Niels Bohr Institute, University of Copenhagen, 2100 Copenhagen, Denmark. [6] These authors contributed equally: Huixia Ren, Yanjun Li, Chengsheng Han. ✉email: lychen@pku.edu.cn; tangc@pku.edu.cn

To precisely regulate the blood glucose level[1-3], glucose elevation induces Ca$^{2+}$ oscillations in pancreatic islet cells that trigger the pulsatile secretion of insulin and glucagon[4-7]. The oscillatory activity of pancreatic islets is widespread in many species, such as mouse[8] and human[5]. Dampening and disappearance of hormone pulsatility and islet Ca$^{2+}$ oscillations are associated with the pathogenesis of diabetes[9-12]. Multiple types of glucose-stimulated oscillation patterns have been observed in islets, including fast (~20 s cycle), slow (>100 s cycle), and mixed oscillations (20~300 s cycle)[13-15]. While different mathematical models have been proposed for the underlying mechanism[16-21], most focused on the intrinsic properties of single or coupled β cells, such as the endoplasmic reticulum Ca$^{2+}$ buffering capacity[19] and the slow metabolic cycle of ATP/ADP ratio during glucose stimulation[18]. These β cell-centric models, however, may not fully explain the observed variety of oscillation patterns in islets. In isolated β cells, slow Ca$^{2+}$ oscillations are usually seen[22-25], while adding glucagon accelerates the Ca$^{2+}$ oscillations[26]. The islet is a micro-organ in which multiple cell types closely interact. The α and β cells show highly correlated Ca$^{2+}$ oscillation patterns[27], and the periodic release of insulin and glucagon is temporally coupled both in vitro[4,28] and in vivo[6,7,29,30]. Furthermore, glucagon enhances insulin secretion and assists in lowering the blood glucose level[31,32]. Thus the extensive autocrine and paracrine interactions between α and β cells[31,33-37] may modulate or even dictate the islet oscillation modes.

The challenge of testing such a hypothesis lies in resolving the identity of individual cells and monitoring their activity in live islets simultaneously. In addition, because the spatial organization of α and β cells are highly heterogeneous from islet to islet[38,39], a quantitative comparison of Ca$^{2+}$ oscillations in different islets is necessary. To address these problems, we have developed a microfluidic device attached to a spinning-disc confocal microscope, which allowed individual cells to be imaged under physiological conditions for up to ~2 hrs. Through the long-term imaging of islets undergoing repeated glucose stimulation, we found that the oscillation mode represents intrinsic islet properties. By constructing a new transgenic mouse line, we could identify the cell types in live islets accurately. Quantitative analysis revealed generic features and quantitative relationships in the oscillation patterns across many islets. In particular, we found that oscillations of the islet α and β cells are synchronized but phase-shifted, and the α cell abundance in the islet correlates with its oscillation mode. Finally, we developed a coarse-grained mathematical model incorporating paracrine interactions between α and β cells. The model reproduced key quantitative features of the experimentally observed oscillations and suggested that different oscillation modes may come from the varied paracrine controls.

## Results

**Glucose-evoked Ca$^{2+}$ oscillations represent an intrinsic property of the islet.** To provide a stable and controllable environment for long-term imaging of intact islets, we developed a microfluidic device (Fig. 1a). On one side, we designed an inlet port to load the islet (300 μm in width and 270 μm in height), which could be sealed after loading. The chip could trap islets of different sizes with a descending PDMS ceiling (270, 180, 150, 110, 80, and 50 μm in height). On the side opposite the inlet, five independent input channels merged into one channel upstream of the islet trapping site. Such a device enabled long-term and stable imaging of islets even during the switching of different perfusion solutions. For instance, when glucose concentration in the perfusion solution was increased from 3 to 10 mM (3 G to

10 G), all islets exhibited an initial rapid rise in cytosolic Ca$^{2+}$, followed by a gradual appearance of fast (cycle < 60 s, 31 of 46 islets from 8 mice in 4 independent isolations), slow (cycle > 100 s, 9 of 46 islets) or mixed (6 of 46 islets) Ca$^{2+}$ oscillations (Fig. 1b). While different islets displayed largely variable Ca$^{2+}$ oscillations, the second round of 10 G stimulation in the same islet evoked an oscillation frequency nearly identical to the first round (Fig. 1c). The spatial activation profiles were also similar, as almost identical cells lighted up at the designated times of Ca$^{2+}$ cycles during the two rounds of stimulation (Fig. 1d). Quantitatively, sequential activations of islet cells between the two rounds of stimulation showed a significantly higher similarity index than random association (Fig. 1e, See Methods). Therefore, specific oscillation modes represent a robust intrinsic property of individual islets, possibly determined by the islet's cell type composition and their spatial organization.

**Identification of islet α and β cell types using transgenic mice.** To probe the specific micro-organization of an islet and to distinguish the Ca$^{2+}$ activities between α and β cells, we generated *Glu-Cre*+; *GCaMP6f*$^{f/+}$; *Ins2-RCaMP1.07* mice in which α and β cells were labeled with the green and red fluorescent Ca$^{2+}$ sensor, GCaMP6f and RCaMP1.07, respectively (Fig. 2a, see Methods). Because the vector was randomly inserted into the genome using the PiggyBac transposon system, β cells were sparsely labeled (9.4%, Fig. 2b and Supplementary Fig. 1). RCaMP1.07 is a Ca$^{2+}$ sensor with a fluorescence on-rate similar to GCaMP6f[40] and Cal-520 AM (Supplementary Fig. 2c and d). We confirmed the labeling accuracy by immunofluorescence. The RCaMP1.07 expressing islet cells were 98.2% insulin-positive, while the GCaMP6f expressing islet cells were 98.3% glucagon positive (Fig. 2c). This result was also confirmed in intact islets using immunofluorescence labeling (Supplementary Fig. 1c and d) and pharmacology experiments (Supplementary Fig. 2a and b). The GCaMP6f expressing cells responded to both NE and glutamate stimulation, while the RCaMP1.07 expressing cells were silent under both stimuli. These data both reinforced the expression specificity of α and β cells, and non-detectable overlaps in emission spectrums between GCaMP6f (525/50 nm) and RCaMP1.07 (600/50 nm) (Fig. 2d and Supplementary Fig. 2e).

Under 3 mM glucose stimulation, all β cells and some α cells remained silent while some α cells demonstrated variable Ca$^{2+}$ transients (Fig. 2e, Movie 1), agreeing with a previous report[27]. Therefore, while the mean Ca$^{2+}$ concentration of β cells was low, a fluctuating elevated mean Ca$^{2+}$ was observed in α cells. Upon 10 mM glucose stimulation, β cells responded with a large initial rise in cytosolic Ca$^{2+}$ followed by slow decay. In contrast, α cells demonstrated two opposite Ca$^{2+}$ responses (Fig. 2f): within the first five minutes after the 10 G stimulation, a minor group of α cells (16% of 267 α cells, 5 islets from 3 mice) demonstrated a pronounced and slow Ca$^{2+}$ transient that was delayed to that of the β cells (we name these α cells as the excited α cells), while the majority of α cells (84%) did not show the initial peak (named as the inhibited α cells). Although this minor population of excited α cells did not fit the consensus, glucose-stimulated Ca$^{2+}$ responses in some α cells were also noted previously[41,42]. In addition, some α cells remained silent both at the 3 G and 10 G conditions (Fig. 2h). All the α cells responded to 25 mM KCl. Therefore, there exist at least three types of α cells in intact islets.

**Islet α and β cells are globally phase-locked.** At the later stage of 10 G stimulation, all glucose-responsive α cells in the islet became synchronized, accompanied by the synchronized Ca$^{2+}$ oscillation of β cells (Fig. 2e and g, Fig. 3a and b, Movie 2). Unlike β cells which are interconnected by the gap junction protein Connexin36

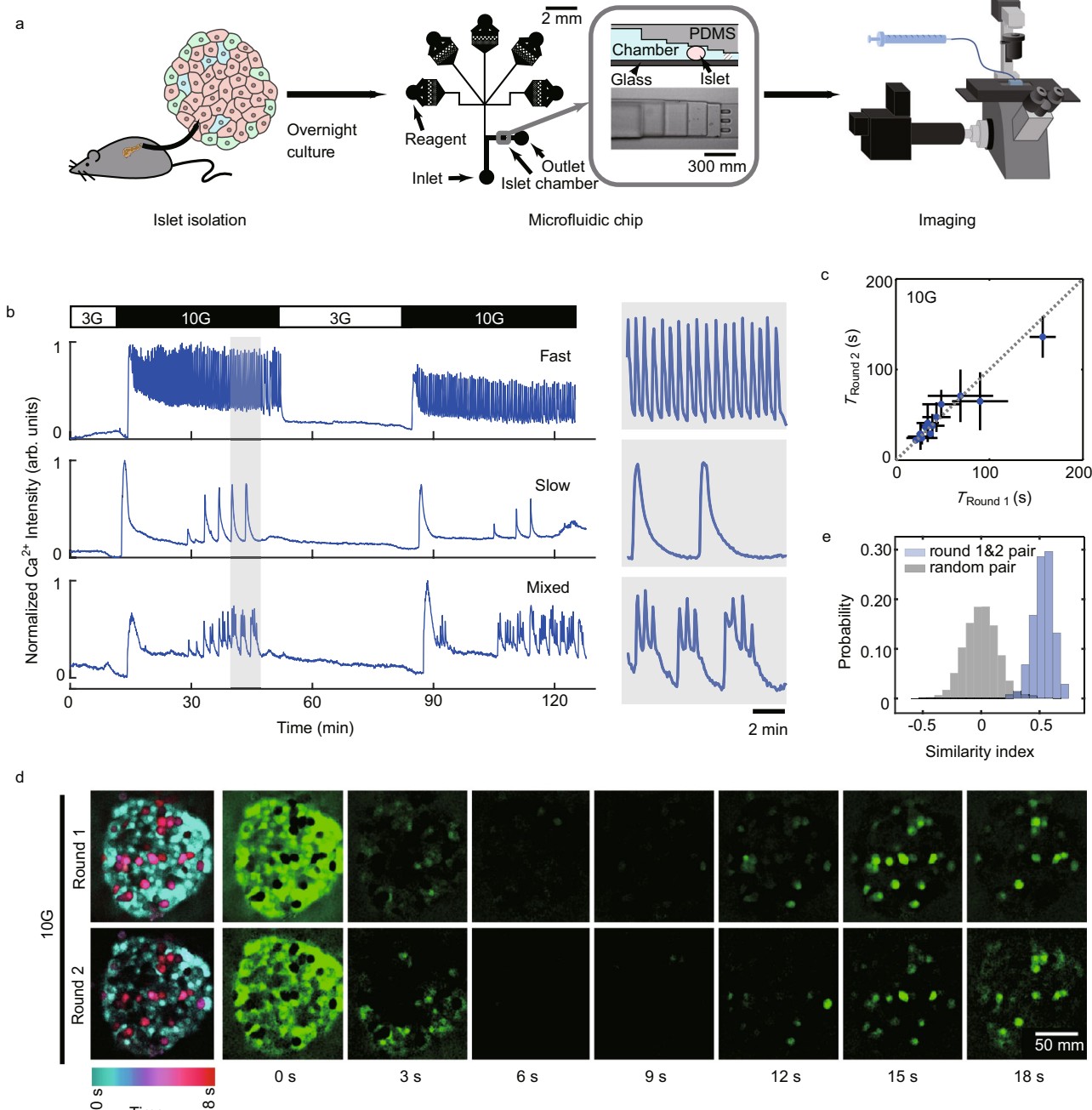

**Fig. 1 Islets Show Intrinsic Ca$^{2+}$ Oscillation Modes under High Glucose Stimulation. a** Experimental flow chart. Islets are isolated from mice. After overnight culture, the islets are loaded onto the microfluidic chip for imaging with spinning-disk confocal microscopy. The chip comprises five reagent channels, an inlet channel and an outlet channel. The islet chamber traps the islet with a gradient height from 270 μm to 50 μm. **b** Representative recordings of whole islet Ca$^{2+}$ signal in Cal-520 AM loaded islets isolated from C57BL/6 J mouse. The islet is stimulated with a repeated protocol: 10 min 3 mM glucose (3 G), 40 min 10 mM glucose (10 G), 30 min 3 G and 40 min 10 G. The first panel, the islet displays fast oscillations with a period of ~20 s (31 of 46 islets); the second panel, slow oscillations at ~3.5 min (9 of 46 islets); the third panel, mixed oscillations at ~20 s and 2.45 min (6 of 46 islets); Enlarged images of the shaded region are shown on the right. **c** The mean Ca$^{2+}$ oscillation period during the first *versus* the second round of 10 G stimulation ($n = 14$ islets). Bars represent mean ± s.d. (standard deviation). **d** Cell activation sequence in the first and second round of 10 G stimulation. We subtract the previous frame from the following frame of the original Ca$^{2+}$ images (frame interval 3 s). Shown is the cell activation sequence averaged across oscillation cycles in a 5 min interval (aligned with the maximum activation frame). The left panels summarize the time sequence shown in the 7 right panels, with the pseudo-color representing the activation time. **e** The same islet shows a high similarity index between the first and the second round of 10 G stimulation (5 islets from 3 mice in 3 independent isolations).

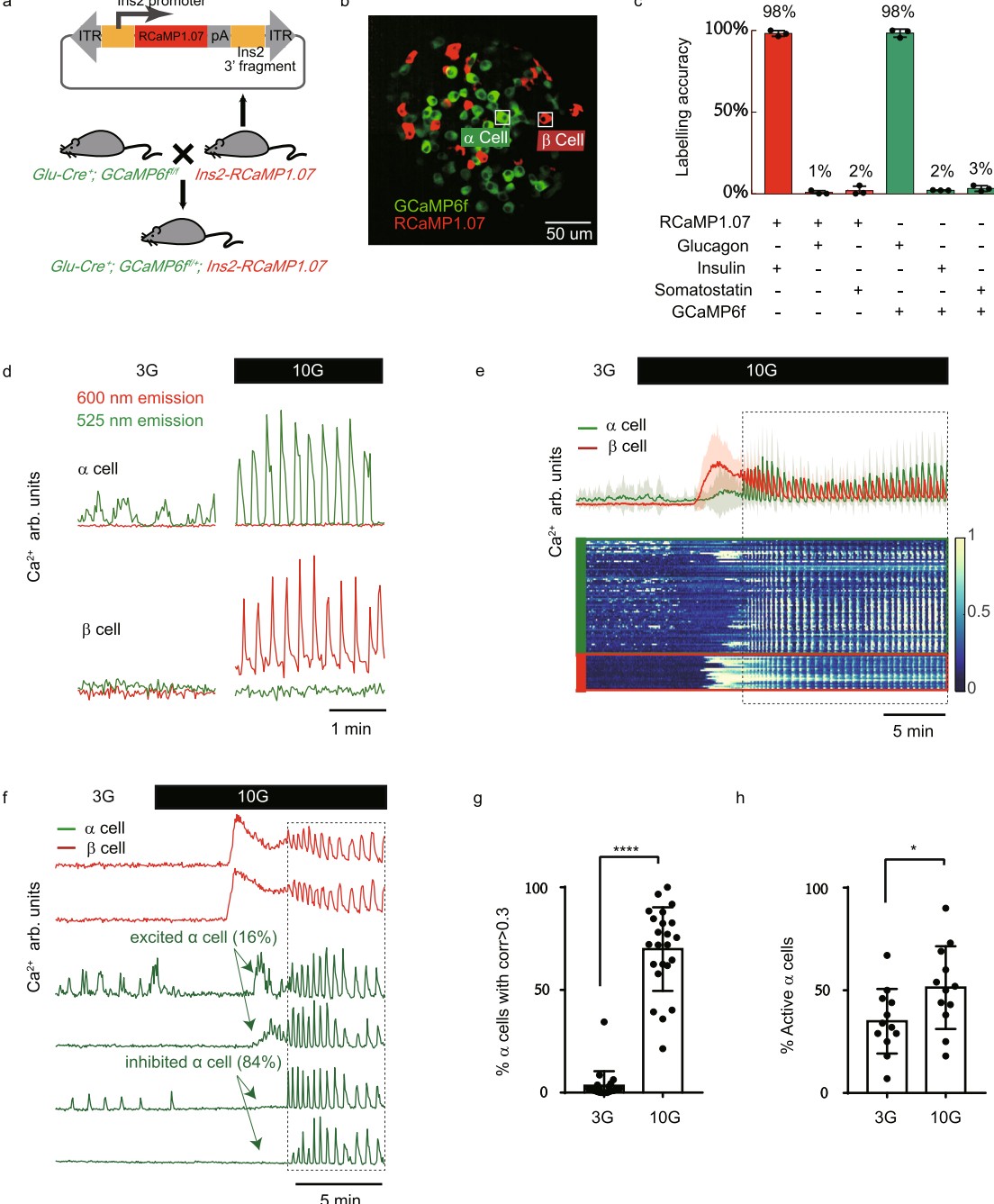

**Fig. 2 Using *Glu-Cre*+; *GCaMP6f*f/+; *Ins2-RCaMP1.07* transgenic mice to identify islet α and β cells. a** Gene targeting vector designed with *Ins2* 5′-promoter, *RCaMP1.07*, and *Ins2* 3′-fragment for constructing the *Ins2-RCaMP1.07* mice. **b** Maximal projection of Ca²⁺ activity from the *Glu-Cre*+; *GCaMP6f*f/+; *Ins2-RCaMP1.07* mice islet. α cells expressed GCaMP6f (Green) and β cells sparsely expressed RCaMP1.07 (Red) (see Methods). **c** Immunofluorescence co-localization analysis of *Ins2-RCaMP1.07* and *Glu-Cre*+; *GCaMP6f*f/f islet cells. $n = 598$, 461 and 505 RCaMP1.07+ cells for insulin, glucagon and somatostatin (3 mice). $n = 939$, 553 and 995 GCaMP6f cells for glucagon, insulin and somatostatin (3 mice). Bars represent mean ± s.d. **d** 525 nm and 600 nm emission (single bandpass filter with width 50 nm) signals from single α and β cells under 3 G (2.5 min) and 10 G (2.5 min) stimulation. The cell positions are marked in **b**. **e** Mean α (green) and β (red) cells Ca²⁺ signal from intact mouse islets exposed to 3–10 mM glucose. Shading corresponds to s.d. The lower panel is the heat map of the normalized single-cell Ca²⁺ signal. Dashed Box shows the stable oscillatory phase. **f** Single α (green) and β (red) cell Ca²⁺ signals from intact mouse islets exposed to 3–10 mM glucose. Third and fourth rows: Excited α cell Ca²⁺ signals (3 G active and inactive). Fifth and sixth rows: Inhibited α cell Ca²⁺ signals (3 G active and inactive). Dashed Box shows the stable oscillatory phase. **g** Percentage of synchronized α cells (mean Pearson correlation coefficient >0.3) under 3 G and 10 G stimulations ($n = 23$ islets from 5 mice in 5 independent isolations). Bars represent mean ± SEM (standard error of the mean). **h** Percentage of active α cells (normalized to 25 mM KCl stimulated α cell number) under 3 G and 10 G stimulations ($n = 12$ islets from 4 mice in 4 independent isolations). Bars represent mean ± SEM.

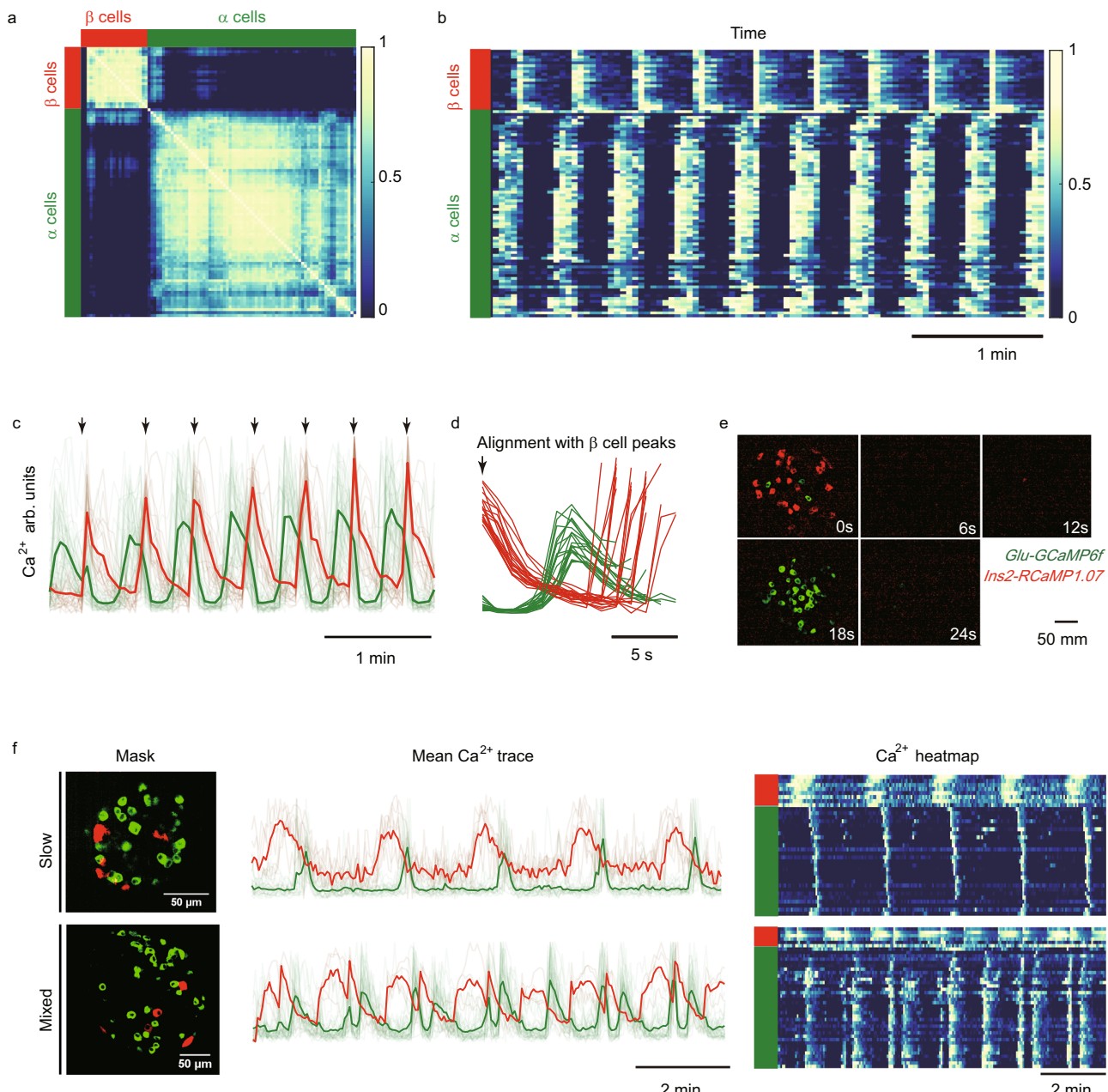

**Fig. 3 α and β Cells are globally phase-locked during oscillation. a** Correlation matrix (Pearson correlation coefficients) for Ca²⁺ activity of *Glu-Cre⁺;* *GCaMP6f^f/+;Ins2-RCaMP1.07* islet cells under 10 G stimulation. Cells are sorted according to cell types. β cells are indicated by the red bar, and α cells by the green bar. **b** The heat-map of time-dependent Ca²⁺ activity for α and β cells under 10 G stimulation. The color bar codes the normalized Ca²⁺ intensity. Each row represents the same cell in **a**. **c** Mean Ca²⁺ activity of α and β cells under 10 G stimulation for the same islet as **a**. Single-cell traces are shown with light lines. The red trace represents β cells and green trace α cells (*n* = 22 for β cells and *n* = 71 for α cells). **d** Mean Ca²⁺ activity of α and β cells in **c**, aligned at each β cell peak. Each trace starts from a peak of β cell oscillation and stops in the next peak. **e** Sequential activation of α and β cells under 10 G stimulation. We subtract the previous frame from the next frame of the original Ca²⁺ images and average across oscillations (each oscillation is aligned with the maximum activation frame). The mean activation sequence uses 5 min Ca²⁺ images (frame interval is 3 s). The β cells are colored red and α cells green. **f** Representative recordings of slow and mixed oscillations of Ca²⁺ activity of α and β cells under 10 G stimulation. Top: Slow oscillation; bottom: Mixed oscillation. The first column, maximal projection of Ca²⁺ masks (α cells green and β cells red); second column, mean Ca²⁺ trace of α and β cells (single-cell traces are shown with light lines); third column, heat-map of α and β cells' Ca²⁺ activity. The color bar codes the normalized Ca²⁺ intensity.

to achieve synchronization[43–47], α cells do not express gap junction proteins and are thus not physically connected[48]. Because mean Ca²⁺ peaks of α cells displayed a fixed delay to those of β cells (~20 s), we hypothesized that the highly synchronous α cell activity might be due to stable phase-locking to the β cell activity (Fig. 3c and d). Consistently we observed global α cell activation after the turning-off of β cells (Fig. 3e). Intriguingly, phase-locking

with similar temporal characteristics was also present in slow and mixed oscillations (Fig. 3f, Movies 3 and 4), suggesting a common underlying mechanism.

**Variable delay of β cell activation after α cell determines the oscillation mode.** To non-biasedly sort out critical parameters

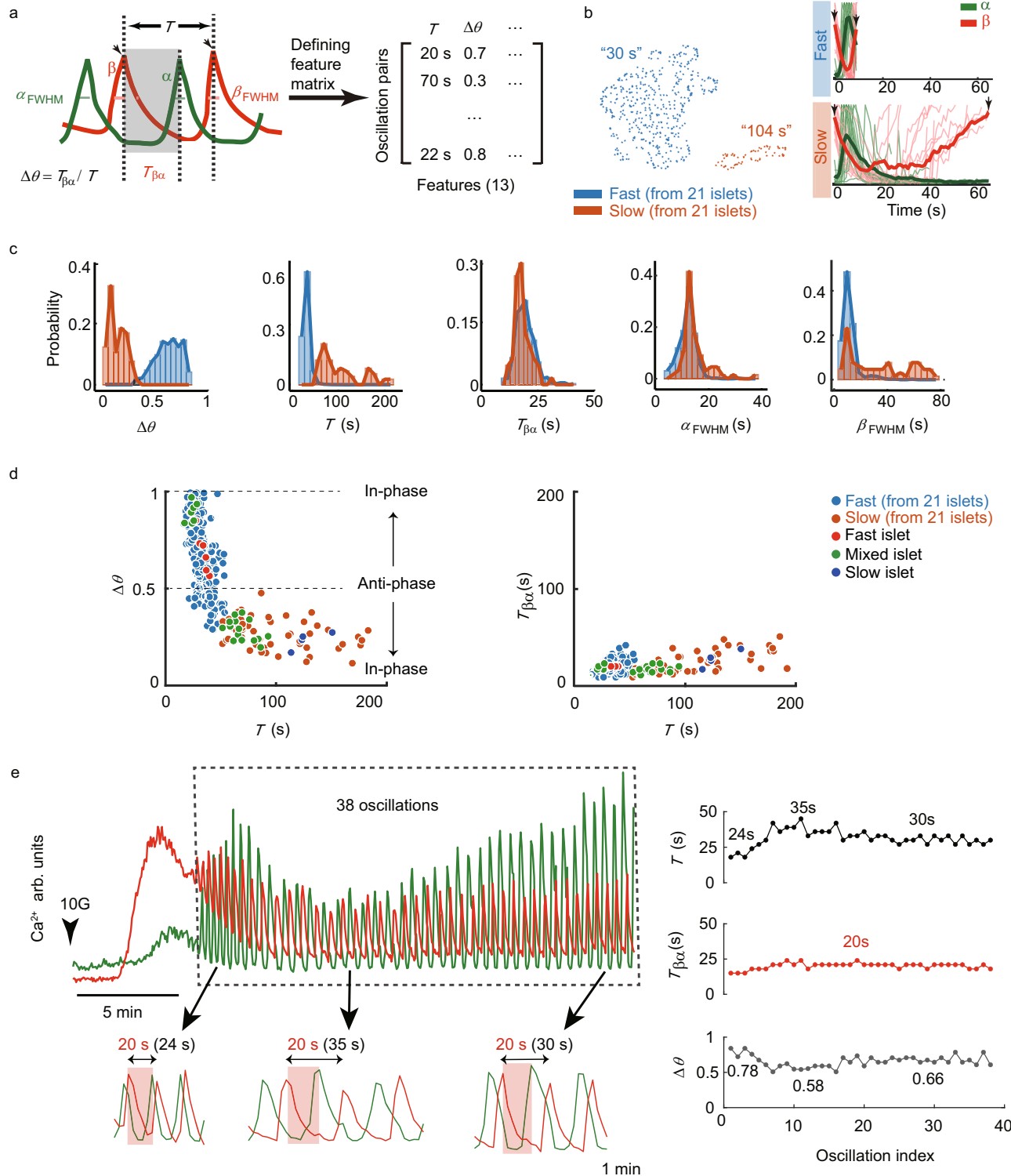

differentiating various $Ca^{2+}$ oscillation patterns, we quantitatively defined features of each oscillation cycle of alternately activated α and β cells (Fig. 4a and Supplementary Fig. 3). The oscillation period $T$ was defined as the time window between one cycle of β cell activation. The waiting time for α cell to activate after β cell activation was defined as $T_{\beta\alpha}$. The phase difference ($\Delta\theta$) of two types of cells was defined as the ratio between $T_{\beta\alpha}$ and $T$. There were multiple ways to determine these quantities—either using the peak, 25%, 50%, or 75% decrease—and they did not change the outcome of the analysis (Supplementary Fig. 4b–d). In total,

we defined 13 features that recapitulated different characteristics of the α and β cell $Ca^{2+}$ dynamics, such as the rise and decay time of the activation for α and β cells, the Full Width at Half Maximum (FWHM) (Table 1, and see Methods for detailed definitions and Source Data). Based on these parameters, the oscillating cycles were vectorized and assembled into a feature matrix. Next, by dimension reduction of these parameters with the UMAP algorithm[49], oscillations could be classified into fast and slow ones (Fig. 4b), in which $T$, $\Delta\theta$ and FWHM of β cells demonstrated significant inter-group differences (Fig. 4c and

**Fig. 4 α Cell activates after β cell with a fixed time delay while β cell activates after α cell with variable time delay. a** Left: Each oscillation cycle is defined as the interval between two β cell activations (see Supplementary Fig. 3 for alternative ways to define the period $T$). Right: 13 features of α-β oscillation cycle are used to construct the feature matrix (columns represent features, rows represent oscillation cycles). **b** Left: Oscillation cycles are separated into two clusters by using UMAP, with the mean oscillation period ($T$) of each cluster shown ($n = 658$ cycles from 21 islets, the fast and slow clusters have 574 and 84 oscillation cycles, respectively). Right: Mean $Ca^{2+}$ traces (bold line) and 15 representative traces (thin line) of fast clusters (top panel) and slow ones (bottom panel), aligned at the peaks of β cell $Ca^{2+}$ activity (arrows). **c** Histogram of phase difference ($\Delta\theta$), oscillation period ($T$), waiting time of α cells ($T_{\beta\alpha}$), and Full Width at Half Maximum of α and β cells ($\alpha_{FWHM}$, $\beta_{FWHM}$) in fast and slow clusters defined in Fig. S3. **d** Left: Scatter plot of $\Delta\theta$ versus $T$. Dashed lines indicate in-phase and anti-phase α and β oscillations. Right: Scatter plot of $T_{\beta\alpha}$ versus $T$. The blue and orange dots represent the fast and slow clusters from all 21 islets in **b**. The red, green, and dark blue dots represent the oscillations from fast, mixed, and slow islets shown in Fig. 5d (right panels) (See Supplementary Fig. 4b–d for scatter plots of other ways to define the period). **e** Left: Mean $Ca^{2+}$ traces of α and β cells when the glucose stimulation was shifted from 3 G to 10 G. The rectangle box showed 38 α and β phase-locked oscillations. Right: $T$, $T_{\beta\alpha}$, and $\Delta\theta$ of 38 oscillation cycles.

**Table 1 The mean value of the features in the fast and slow clusters.**

|  | Fast cluster | Slow cluster | P value | Significance |
|---|---|---|---|---|
| $T$ (s) | 29.8 | 99.9 | 1.487e-161 | **** |
| $\Delta\theta$ | 0.7 | 0.2 | 3.259e-99 | **** |
| $T_{\beta\alpha}$ (s) | 23.3 | 19.5 | 8.59e-5 | n.s. |
| $\alpha_{FWHM}$ (s) | 11.9 | 14.8 | 6.573e-13 | n.s. |
| $\beta_{FWHM}$ (s) | 11.4 | 31.1 | 1.175e-53 | **** |
| $\alpha_{FWHM}$ / $\beta_{FWHM}$ | 1.3 | 0.6 | 5.98e-10 | n.s. |
| $\tau_{\alpha\_up}$ (s) | 5.05 | 5.0 | 0.7799 | n.s. |
| $\tau_{\alpha\_decay}$ (s) | 8.8 | 13.6 | 1.31e-23 | * |
| $\tau_{\beta\_up}$ (s) | 5.1 | 18.7 | 7.46e-38 | ** |
| $\tau_{\beta\_decay}$ (s) | 10.4 | 20.4 | 3.08e-60 | **** |

Statistical comparisons were conducted using unpaired two-tailed t test. *p <1e-15, **p <1e-30, ***p <1e-40, ****p <1e-50. Sample sizes of fast cluster and slow cluster were 574 pairs and 84 pairs, respectively. For detailed definitions, please see Methods.

Supplementary Fig. 4a). The $T$ for the fast oscillations centered at ~30 s, threefold smaller than the slow ones (~100 s). Similarly, the average $\Delta\theta$ for the rapid oscillations was threefold larger than the slow ones (~0.7 versus ~0.2, Table 1). The mean $T_{\beta\alpha}$ was indistinguishable between the fast and slow oscillations. The mean β cells' FWHM in the fast oscillations was shorter than that of the slow ones (~20 s versus ~23 s) (Fig. 4c and Table 1). In contrast, the distributions of FWHM for α cells between the fast and slow oscillations were similar (Fig. 4c). This implies a higher FWHM ratio of the $Ca^{2+}$ transients of α to β cells in the fast $Ca^{2+}$ oscillations (~1.3) than in the slow ones (~0.6) (Supplementary Fig. 4a).

As $T$, $\Delta\theta$, and $T_{\beta\alpha}$ were inter-dependent parameters, we further evaluated their relationship using scatter plots (Fig. 4d). While $T_{\beta\alpha}$ remained constant (~20 s) across all different period $T$, the phase-shift between α and β cells $\Delta\theta$ decreased with $T$. When the β cells showed little delay (~0 s) to the α cells (Fig. 4d, $T = ~20$ s), the two types of cells were nearly in-phase ($\Delta\theta = ~1$). When the β cell's period was around ~40 s (Fig. 4d, $T = ~40$ s), the α and β cells were nearly anti-phase ($\Delta\theta = ~0.5$). Finally, when the β cells demonstrated a much longer period per cycle ($T = ~200$ s), the α and β cells appeared nearly in phase again. These relationships hold for various oscillation modes, as indicated by the different colors in Fig. 4d.

Interestingly, sometimes we observed that the phase shift between α and β cells underwent an initial transient period during which it changes in time before stabilizing at a steady value. This was usually due to a changing β cell period as the delay of α cell activation following β cell ($T_{\beta\alpha}$) was typically fixed. For example, in Fig. 4e, while $T_{\beta\alpha}$ remained stable at 20 s during the whole process, $T$ started from 24 s, and extended to 35 s in the following

ten oscillations before finally stabilizing at 30 s (Table S1, $n = 3$ islets from 1 mouse). This data corroborated our analysis conducted in multiple islets and pointed to the possibility of dynamic changes in the interactions between α and β cells in the same islet. Because isolated single β cells displayed only slow oscillations with a period of about 6 minutes (Supplementary Fig. 5a and b), we speculate that increased stimulatory effects from α cells to β cells may accelerate β cell oscillation.

**Mathematical modeling.** We observed that islets show heterogeneous yet intrinsic oscillation patterns under high glucose stimulation. To better understand the origin of various oscillation modes and the factors controlling them, we developed a mathematical model incorporating interactions between α and β cells.

Given the fact that both α cells and the β cells in an islet were globally synchronized respectively, we simplified the islet as a model of two coupled "cells" - an α cell and a β cell (Fig. 5a). The oscillation of each cell was described by a phase variable $\theta$, which was driven by an intrinsic force and a paracrine force. The equations are

$$\frac{d\theta_\alpha}{dt} = \omega_\alpha + K_{\beta\alpha}f_s(\theta_\beta)f_{r\alpha}(\theta_\alpha) \tag{1}$$

$$\frac{d\theta_\beta}{dt} = \omega_\beta + K_{\alpha\beta}Gf_{r\beta}(\theta_\beta) \tag{2}$$

$$\frac{dG}{dt} = f_s(\theta_\alpha) - \tau G. \tag{3}$$

The intrinsic term $\omega_\alpha$ and $\omega_\beta$ corresponded to the oscillation frequencies of the single isolated cells. As shown by the previous and our studies, single β cells oscillate for a period of ~3–6 minutes, and single α cells oscillate for a period of ~30–60 s (Supplementary Fig. 5). The paracrine force came from the inhibitory factor (insulin) secreted by the β cell and the stimulatory factor (glucagon) secreted from the α cell. As shown in Eqs. (1)-(3), $G$ represented the glucagon concentration in the system which was determined by α cell secretion $f_s(\theta_\alpha)$ and degradation, $f_s(\theta_\beta)$ represented the β cell secreted insulin concentration, $f_{r\alpha}(\theta)$ represented the paracrine inhibition of α cell by β cell and $f_{r\beta}(\theta)$ represented the paracrine stimulation of β cell by α cell. The main results of the model were insensitive to the choice of the specific forms of $f_s(\theta)$, $f_{r\alpha}(\theta)$, and $f_{r\beta}(\theta)$ as long as they were periodic functions resembling the general characteristic of the biology (see Supplementary Figs. 6 and 7 for details). The coefficients $K_{\alpha\beta}$ and $K_{\beta\alpha}$ represented the coupling strengths between the α and β cells. Note that this is a two-phase model without providing any information on the oscillation amplitudes. Its behavior can be characterized by the "winding number"

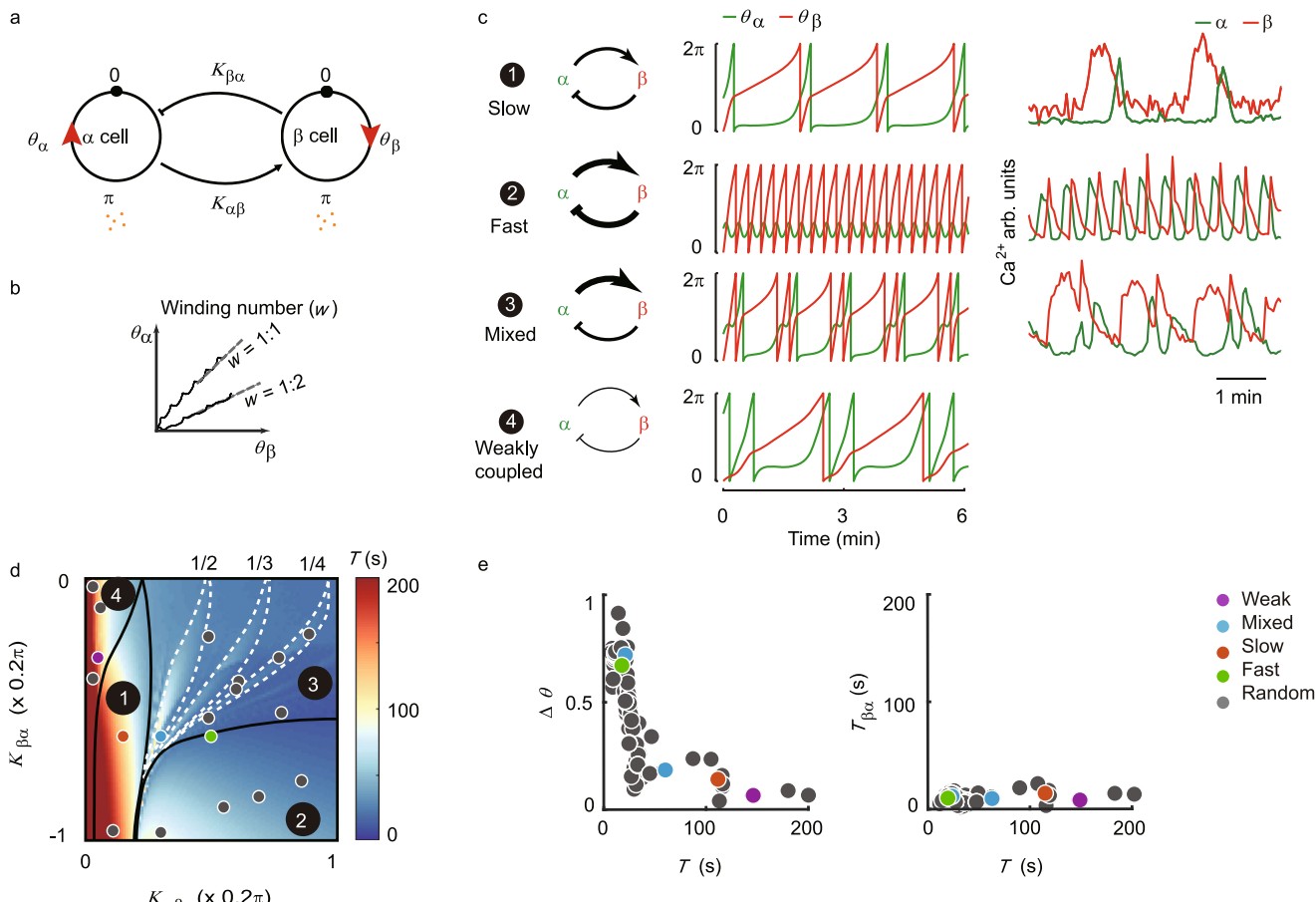

**Fig. 5 Mathematical model of islet α and β cells. a** Two-cell islet model. The state of each cell is described by its phase θ. Cell secretes hormone at phase π. The hormone secreted by α cell stimulates β cell (with strength $K_{αβ}$), and the effect of β cell activation inhibits α cell (with strength $K_{βα}$). **b** Illustration of winding number (w) definition. Two examples are shown. Solid lines are the actual trajectories of two solutions of the equations, and dotted lines are straight lines representing the asymptotic limits of the trajectories that define the winding number. **c** Left panel: Schematic of α and β cells' interaction strengths. Four cases from the four regions in **d** are shown. Middle panel: Example traces of $θ_α$ and $θ_β$ in the four cases, respectively. The parameters used in these examples are indicated by the four colored dots in **d**. Right panel: Corresponding examples of $Ca^{2+}$ traces found experimentally for the first three cases. **d** The phase diagram of the system. Depending on the two coupling constants $K_{αβ}$ and $K_{βα}$, the oscillatory behavior of the two cells falls into one of the four phase-locked regions, characterized by the winding number w. In region 3, any rational winding number w < 1 has a stable phase-locking region. For clarity, only the phase-locking regions for lower-order rational numbers (1/2, 1/3 and 1/4) are shown. Color bar codes the period of the oscillation. Except for the four colored dots, 15 randomly selected gray dots are also shown. **e** Scatter plots of Δθ and $T_{βα}$ versus T. Each dot represents one oscillation cycle. The color of the dot indicates the parameters used in the simulation, which are shown in phase diagram (**d**) with the same color.

defined as the asymptotic ratio of the two phases $w = θ_α/θ_β$ (Fig. 5b).

In the model (1)-(3), the stimulation from α cell to β cell is determined by the accumulation level of the glucagon, while the inhibition from β cell to α cell depends on the instantaneous secretion of the β cell. This is the case when the time scale of α cell oscillation is faster than the degradation/disappearance time of the glucagon, and the inhibitory effect from β cell to α cell is strong, as suggested by the invariant $T_{βα}$. In the supplemental materials, we have analyzed other cases in which both the stimulatory and inhibitory factors depend on the instantaneous secretion of α and β cells (Supplementary Fig. 6g) or both determined by the accumulated level set by the secretion and degradation (Supplementary Fig. 6h). The results were similar, and in particular the phase-locking phenomenon was preserved.

By adjusting the coupling strengths ($K_{αβ}$ and $K_{βα}$) between the α and β cells, our model displayed all three types of oscillation behaviors observed in experiments (cases 1–3, Fig. 5c and Movie 5). When α cell weakly stimulated β cell (small $K_{αβ}$), the model islet generally showed slow oscillations. When α cell and β

cell were strongly coupled with each other (large $K_{βα}$ and $K_{αβ}$), the model islet generally showed fast oscillations. When α cell strongly stimulated β cell and β cell weakly inhibited α cell (large $K_{αβ}$ and small $K_{βα}$), the model islet generally showed mixed oscillations. With very weak coupling between α and β cells, the model displayed an oscillation behavior similar to uncoupled single cells but not islet experiments (Fig. 5c, case 4). Further quantification found the phase difference Δθ and the period T displayed an inverse proportional relationship (Fig. 5e, left panel), which was because of a constant waiting time of the α cell regardless of the oscillation modes (Fig. 5e, right panel), similar to the experimental data (Fig. 4d). Here the coupled oscillator model indicated that the strengths of the paracrine interaction determine the oscillation modes.

We next analyzed the model's behavior by systematically varying the coupling strengths between the α and β cells. We found that the α and β cells were generally phase-locked, i.e., their phases were dependent on each other with a fixed relationship characterized by the winding number w (Fig. 5b). By plotting the winding number's dependence on the coupling strengths, we

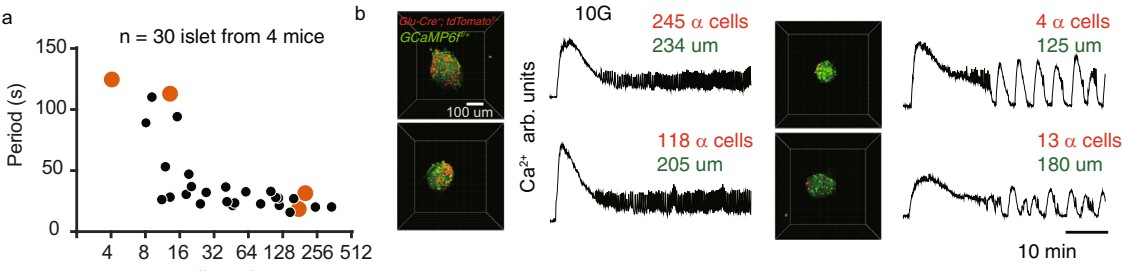

**Fig. 6 Islet oscillation frequency depends on the number of α cells. a** Scatter plot of islet's α cell number (tdTomato positive cell number) and mean islet Ca²⁺ oscillation period under 10 G stimulation ($n = 30$ islets from 4 *Glu-Cre⁺; tdTomato^{f/+}; GCaMP6f⁺* mice in 3 independent isolations). The islets labeled with yellow edge dots were shown in **b**. **b** 3D image of the islets labeled in **a**. First and third columns: The tdTomato was expressed in α cells (red) and GCaMP6f in whole islet cells (green). Second and fourth columns: Mean Ca²⁺ signal with 40 min 10 G stimulation. The number of α cells (tdTomato positive cells) and the diameter of the islet are shown.

obtained the phase space of the system of two coupled oscillators (Fig. 5d). It formed a two-dimensional Arnold tongues[50], which could be separated into four regions (Fig. 5d, see Methods). In region 1, the α cell and the β cell are locked on the $w = 1/1$ mode. That is, when $θ_β$ finishes one cycle ([0,2π]), $θ_α$ will also finish one cycle. An example of this oscillation mode is shown in Fig. 5c (upper panel) and Movie 5 (top left panel). In region 2, the two cells are locked in the mode $w = 0/1$. That is, when $θ_β$ finishes one cycle ([0,2π]), $θ_α$ cannot finish a full cycle before being pushed back (Fig. 5c (middle-upper panel) and Movie 5 (top right panel)). Here the strong stimulation from α to β induces a fast oscillation frequency, while the strong repression from β to α prevents the α cell from finishing a full cycle every time it is activated. In region 3, the two cells are locked with $0 < w < 1$. In particular, there exist $w = m/n$ modes, where $m < n$ are both integers. In a mode with $w = m/n$, when $θ_β$ finishes $n$ cycles ([0,2nπ]), $θ_α$ will finish $m$ cycle(s) ([0, 2mπ]). In the example with $w = 1/2$ shown in Fig. 5c (middle-lower panel) and Movie 5 (bottom left panel), while each activation of the β cell can finish one full cycle, the first activation of α cell cannot finish a full cycle before being pushed back, and only the second activation can finish a full cycle. In region 4, the two phases are locked with $w > 1$, which means that α cell will finish more cycles than β cell. An example of $w = 2/1$ is shown in Fig. 5c (lower panel) and Movie 5 (bottom right panel). In this region, α and β cells couple weakly. At the upper left corner $K_{αβ} = K_{βα} = 0$, α and β cells completely decouple and beat on their intrinsic frequencies. Note that while $w$ jumps discontinuously with continuously varying parameters, the average period of β cell oscillation changes smoothly (Fig. 5d, heat map). Thus, the paracrine interaction between α and β cells offers robust and tunable oscillation patterns and periods.

Our model predicts that the oscillation period critically depends on the α cell's paracrine stimulation. Using two-photon microscopy, we could identify individual α cells in an islet and record the islet's 10 G stimulated Ca²⁺ signal (*Glu-Cre⁺; tdTomato^{f/+}; GCaMP6f⁺* mice islet). We found that the Ca²⁺ activities of islets at 10 G were highly correlated with their α cell numbers (Fig. 6a and b, and Methods). Specifically, the islet's oscillation frequency was positively correlated with the number of α cells in the islet. These data strongly support the model, and show that paracrine contribution from islet α cells is associated with rapid Ca²⁺ oscillation in islet β cells.

Our model further predicts that the oscillation period may be tuned with the strengths of paracrine interaction, depending on the original position of the islet system in the phase space (Fig. 5d, heat map, 7b, Supplementary Figs. 8e and 9f). In particular, increasing the activation from α cell to β cell ($K_{αβ}$)

could decrease the oscillation period, especially in islets of slow oscillations (Fig. 5d, heat map). We applied glucagon (100 nM) to the islets showing fast and slow oscillations (Fig. 7a and Supplementary Fig. 8a). While adding glucagon did not affect fast oscillating islets, it switched islets harboring slow oscillations into fast ones (Fig. 7b–d). Moreover, consistent with the model, the change of oscillation period did not affect the waiting time $T_{βα}$ (Supplementary Fig. 8b–e). On the other hand, the model predicted that reducing the effect of glucagon may lead to more autonomous cellular regulation and slow oscillations (Fig. 7b, f, and Supplementary Fig. 9f). Indeed, by combining insulin and the GCGR and GLP-1R antagonists (MK0893 (MK) and Exendin 9–39 (Ex9)) to inhibit glucagon secretion and its downstream target[31,33], ~50% of the fast oscillatory islets switched to slow oscillations (Fig. 7a). The change of modes reversed back when the inhibitory agents were removed (Supplementary Fig. 9a). Islets' fast-to-slow mode switching relied on the activation level of downstream targets of glucagon. A weaker glucagon receptor antagonist combination prolonged the islet oscillation period without inducing fast-to-slow mode switching (Supplementary Fig. 9b and c). The β-cell-specific GCGR knockout mice (*Ins1-cre;Gcgr^{f/f}*)[51] had fewer fast oscillation islets (Supplementary Fig. 9d). And ~50% of the fast oscillation *Ins1-cre;Gcgr^{f/f}* islets turned into slow oscillations with the weaker glucagon receptor antagonist combination (Supplementary Fig. 9e).

Finally, our model also predicted that in mixed oscillation modes, decay times of Ca²⁺ transients in α cells were different - only the last Ca²⁺ transient in each cluster of cycles of mixed oscillation was independent of β cells, while all other ones were repressed by β cells and should descend faster (Fig. 7g). By analyzing the Ca²⁺ traces of islets with mixed oscillation modes, we confirmed that the decay times of α cell transients fell into two groups: the Ca²⁺ transients in α cells just preceding an uprising of β cell activation decayed faster. In contrast, the ones posterior to β cell transients exhibited a significantly slower decay (Fig. 7h). Overall, the agreement between the model and experiment highlights the importance of α-β interactions in shaping the oscillation modes.

## Discussion

In this study, we developed a microfluidic device that enabled stable and repeatable long-term imaging of islet's Ca²⁺ activity at single-cell resolution. Despite the apparent heterogeneity in Ca²⁺ activity across different islets, individual islets exhibited their own spatial and temporal patterns of Ca²⁺ oscillation that were repeatable under multiple rounds of glucose stimulation. This suggests that the oscillation mode results from intrinsic islet

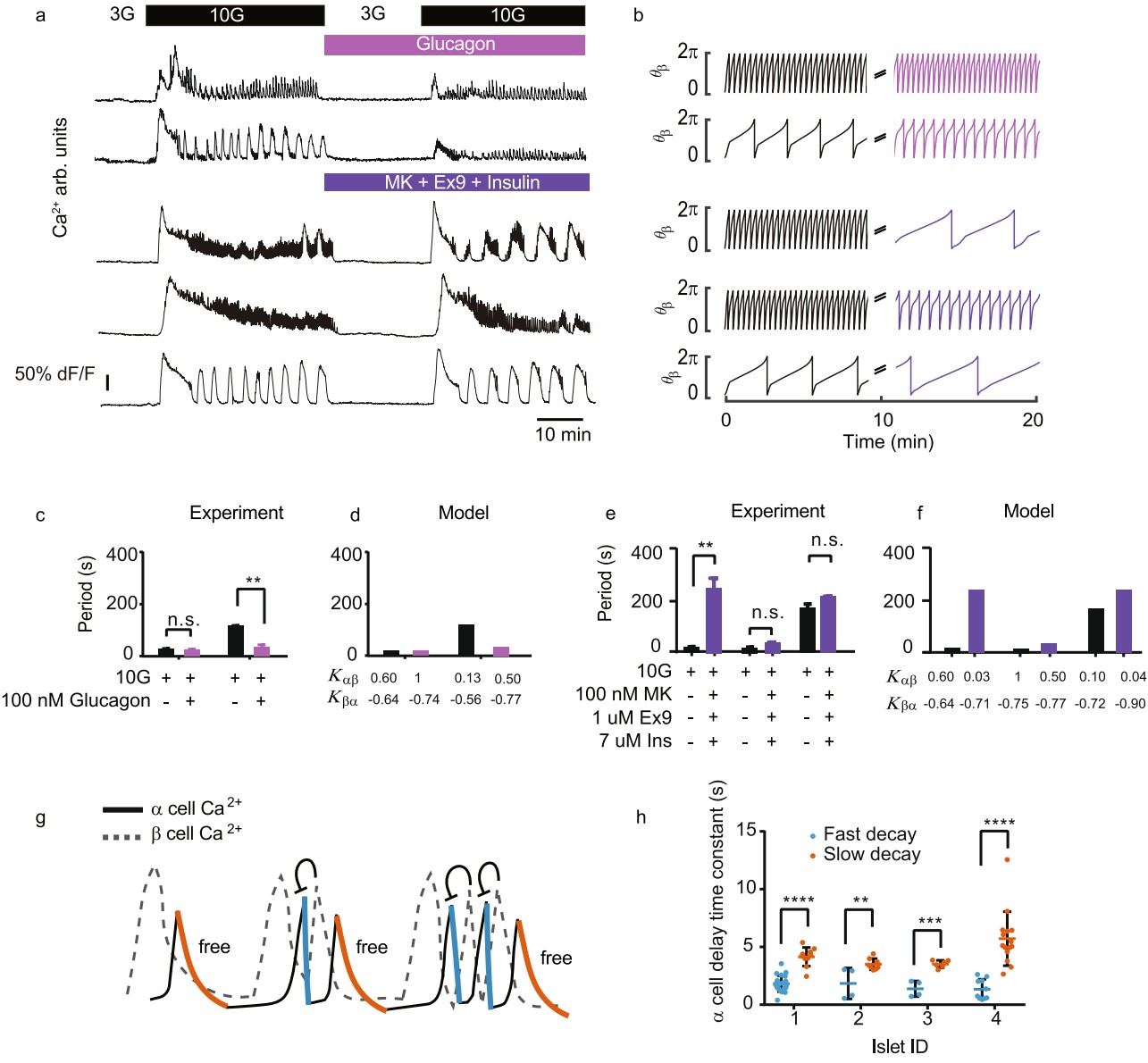

**Fig. 7 Model predictions and experimental support. a** Top two rows: Mean β cell Ca²⁺ signal (*Glu-Cre⁺;GCaMP6f^{f/+};Ins2-RCaMP1.07* islet, see Supplementary Fig. 8a for both α and β signals). The stimulation is 3 G (10 min), 10 G (40 min), 3 G + 100 nM glucagon (20 min) and 10 G + 100 nM glucagon (35 min). Bottom three rows: Mean β cell Ca²⁺ signal (*Ins⁺/⁻;GCaMP6f^{f/+}* islet). The stimulation is 3 G (10 min), 10 G (40 min), 3 G + 100 nM MK (MK0893) + 1 uM Ex9 (Exendin (9–39)) + 7 uM insulin (20 min) and 10 G + 100 nM MK + 1 uM Ex9 + 7 uM insulin (35 min). **b** Modeling parameters in the first row: $K_{\alpha\beta} = 0.60$, $K_{\beta\alpha} = -0.64$ for 0–10 min, and $K_{\alpha\beta} = 1$, $K_{\beta\alpha} = -0.74$ for 10–20 min. Second row: $K_{\alpha\beta} = 0.13$, $K_{\beta\alpha} = -0.56$ for 0–10 min and $K_{\alpha\beta} = 0.5$, $K_{\beta\alpha} = -0.77$ for 10–20 min. Third row: $K_{\alpha\beta} = 0.60$, $K_{\beta\alpha} = -0.64$ for 0–10 min and $K_{\alpha\beta} = 0.03$, $K_{\beta\alpha} = -0.71$ for 10–20 min. Fourth row: $K_{\alpha\beta} = 1$, $K_{\beta\alpha} = -0.75$ for 0–10 min and $K_{\alpha\beta} = 0.50$, $K_{\beta\alpha} = -0.77$ for 10–20 min. Fifth row: $K_{\alpha\beta} = 0.10$, $K_{\beta\alpha} = -0.72$ for 0–10 min and $K_{\alpha\beta} = 0.04$, $K_{\beta\alpha} = -0.90$ for 10–20 min. **c** Mean Ca²⁺ oscillation period (n = 4 fast-to-fast and 3 slow-to-fast islets from 3 mice in 3 independent isolations). Bars represent mean ± s.d. **d** Mean period from the model in the top two rows of **b**. **e** Mean Ca²⁺ oscillation period (n = 5 fast-to-slow, 4 fast-to-fast and 5 slow-to-slow islets from 4 mice in 4 independent isolations). Bars represent mean ± s.d. **f** Mean period from the model in the bottom three rows of **b**. **g** Schematic of α (solid) and β (dashed) cell mix Ca²⁺ trace in islet. The fast (orange) and slow (blue) α cell transients. **h** The decay time constants (see Supplementary Fig. 3 for the definition) for the fast (orange) and slow (blue) α cell transients described in **g** in mixed oscillations (4 islets). Bars represent mean ± SEM.

properties, possibly related to the relative composition of different cell types and their spatial distribution in the islet.

Using the *Glu-Cre⁺; GCaMP6f^{f/+}; Ins2-RCaMP1.07* transgenic mice, we discovered that the α and β cells were globally phase-locked to various oscillation modes. The PiggyBac approach led to the sparse labeling of β cells—it enabled a clear separation of α and β cell Ca²⁺ dynamics. Besides α and β cells, pancreatic islet δ cells have recently received increasing attention in glucose

regulation[52], which are not included in the current model. δ cells are connected to β cells by gap junctions and exhibit Ca²⁺ activities similar to β cells[53]. It releases somatostatin to inhibit both α and β cells strongly. Further study about the δ cell Ca²⁺ dynamics and simultaneous α, β, and δ cell Ca²⁺ imaging would be essential to understand the role of δ cell in tuning α and β phase-locking patterns. In the current mathematical model, the role of δ cells in regulating the oscillation modes was lump-

summed together with that of β cells. It is essential to differentiate the two types of cells in future modeling.

A key finding in our study is that the time delay for α cells' activation following the activation of β cells ($T_{\beta\alpha}$) was invariant, regardless of the islet-to-islet variations in oscillation frequencies and modes. This observation of invariant $T_{\beta\alpha}$ echoed nicely with the ~20 seconds recovery time of α cells from the relief of opto-genetic activation of β and δ cells[53]. Therefore, $T_{\beta\alpha}$ is likely to be determined by one or more secreted factors released by β and δ cells, including insulin, $Zn^{2+}$, ATP, GABA, and somatostatin[47].

In contrast, what varied in different islets under different conditions was the β cells' period ($T$). Our and previous studies suggest that glucagon is a contributing factor in tuning the oscillation mode[14]. Glucagon may elevate cytosolic $Ca^{2+}$ concentration and increase oscillation frequency, both through cAMP-dependent[54,55] and independent pathways[56–59]. Previous studies found that glucagon could speed up fast islets[60], but here we found the acceleration effects were mainly limited to slow islets. On the other hand, when we applied a combination of insulin and glucagon/GLP-1 receptor antagonists to fast-oscillating islets, the switch to slow oscillation was only partial. Recently we have shown that using a combination of glucagon/GLP-1 receptor antagonists failed to suppress islet insulin release to the same extent as glucagon KO and to antagonize GLP-1 receptor[51]. This suggests that it may be difficult to abolish paracrine effects within the closely-packed islets efficiently.

Our work found that α and β cells were both activated under high glucose stimulation. This suggests that the classic picture - α cells secrete glucagon to raise glucose level when it is low and β cells secrete insulin to lower glucose level when it is high – may be over simplified. Previous studies have found that at high glucose concentrations, insulin and glucagon both oscillate[4–7] and that glucagon promotes insulin secretion[26,27,31,41,51,61]. Our work sheds further light on how α and β cells collaborate under high glucose.

In our study, both experimental observation and model simulation showed a robust phase-locking phenomenon between α and β cells. It is known that faster islet $Ca^{2+}$ oscillations display more regular oscillation patterns[14,62]. In light of our findings, the increased regularity may come from the increased stability in phase-locking: more rapid oscillation implies more robust activation from α to β cells and thus a tighter regulation. The $Ca^{2+}$ oscillation of the two types of cells can phase-lock to a variety of modes determined by the paracrine interactions, which not only have different oscillation frequencies, but also display a range of other quantitative features such as the winding number and the ratio of the half-widths for α and β transients. Phase-locking to different frequencies and modes could ensure a stable and tunable secretion of insulin and glucagon. A variety of β cell $Ca^{2+}$ oscillation modes were observed in vivo, including those of fast ones[63–65]. Further studies combining islet $Ca^{2+}$ imaging with real-time detection of α and β cell secretion are needed to investigate the physiological role of the phase-locking and its dependency on paracrine interactions.

Previous studies tried to explain the distinct $Ca^{2+}$ oscillation modes in intact islets with single beta-cell models, which did not consider the contribution from other cell types in islets[16–19]. They typically show much slower oscillation (5–10 min), closer to the oscillation period of isolated single β cells[25,27,66]. Our model emphasized how paracrine interactions may play essential roles in various islet $Ca^{2+}$ oscillation modes. We note that the oscillation period of β cells can be influenced by many factors, such as BK K(Ca) channel blockage[67,68], enhanced ER calcium efflux[69], and the recently discovered pyruvate kinase activation[70]. It is conceivable that both intrinsic properties of single cells and interactions among different cell types may contribute to the regulation of islet $Ca^{2+}$ oscillation modes. Further investigation using pseudo-islets with varying compositions of α and β cells may help to differentiate the roles of intrinsic and paracrine contributions[71].

## Methods

**Animal studies**. The study was conducted according to the guidelines of the Declaration of Helsinki and approved by the Ethics Committee of Peking University (protocol code #IMM-ChenLY-1). All animal experiments were conducted following the institutional guidelines for experimental animals at Peking University, which were approved by the AAALAC. Mice were housed with a 12-h on/12-h off light cycle, 20–24 °C ambient temperature and 40–70% humidity. To generate *Ins2-RCaMP1.07* mice, the targeting vector was constructed with *Ins2* 5'promoter, *RCaMP1.07*, and *Ins2* 3'fragment as described in Fig. 2a. The vector was then injected with PiggyBac mRNA to the zygotes of C57BL/6 J mice (conducted by the Model Animal Research Center of Nanjing University, Nanjing, China). The vector was randomly inserted to the genome using the PiggyBac transposons system, and the β cells were sparsely labeled with RCaMP1.07. *Glu-Cre⁺; GCaMP6f^{f/f}* mice were generated by cross-breeding *Glu-Cre* mice[72] with *Rosa26-GCaMP6f^{flox}* mice (Jackson Laboratory, Strain #:028865). By crossbreeding *Ins2-RCaMP1.07* mice with *Glu-Cre⁺; GCaMP6f^{f/+}* mice, both α and β cells were labeled by different fluorescent $Ca^{2+}$ indicators. C57BL/6 J mice (male) were purchased from the Model Animal Research Center of Nanjing University, Nanjing, China. *Ins1-Cre⁺; GCaMP6f^{f/f}* mice were generated by crossbreeding *Ins1-Cre* mice (Jackson Laboratory, Strain #:026801) with *Rosa26-GCaMP6f^{flox}* mice. *Glu-Cre⁺; tdTomato^{f/f}* mice were generated by crossbreeding *Glu-Cre* mice with *Rosa26-tdTomato^{flox}* mice (Jackson Laboratory, Strain #:007909). *Ella-Cre⁺;GCaMP6f^{f/+}* mice were generated by multiply rounds crossbreeding *EIIa-Cre* mice (Jackson Laboratory, Strain #:003314) with *Rosa26-GCaMP6f^{flox}* mice. *Glu-Cre* mice were isolated from the *Glu-Cre⁺;EYFP^{f/+}* mice provided by Herbert Y. Gaisano Lab of the University of Toronto (Toronto, Canada). *Rosa26-tdTomato^{flox}* mice were isolated from the *Sst-Cre⁺/⁻;tdTomato^{f/+}* mice[73] provided by Xiao Yu Lab of Shandong University (Shandong, China). *Rosa26-GCaMP6f^{flox}* mice, *Ins1-Cre* and *EIIa-Cre* mice were obtained from Jackson Laboratory. *Ins1-Cre;Gcgr^{f/f}* mice had been reported previously[51]. All mice were genotyped by PCR using template tail DNA extracted by the TIANamp Genomic DNAKit (DP304, TIANGEN). *Ins2-RCaMP1.07* mice were PCR genotyped with PrimeSTAR® GXL DNA Polymerase (R050A, Takara) using primers (forward 5'-TGTTTCCGGTCTGACTCTGATTCC-3', reverse 5'- GACCGGCCTTATTCCACTTACGAC-3'). Other mice were PCR genotyped with 2x EasyTaq® PCR SuperMix (AS111, TRANSGEN). Genotyping primers and protocols have been reported previously[72,73] or are available at Jackson Laboratory's website. Mice were maintained in one cage with a light/dark cycle of 12 h and administered chow diet ad libitum. Only male mice aged 8–20 weeks were used in the study.

**Design and application of microfluidic chip**. Microfluidic chips were fabricated by using the elastomer polydimethylsiloxane(PDMS)[74]. Briefly, we used the photo-polymerizable epoxy resin (SU-8-2100) to make a positive relief master, and the PDMS mold was cured on the master. PDMS mold was removed from the master as the channeled substrate. Then we used an oxygen plasma treatment to bond the PDMS mold with a glass coverslip (24 × 60 mm), as shown in detail in Fig. 1a. The islet trapping region was designed as a stair-like channel, using six different thicknesses of SU-8 photoresist. To trap islets of different sizes, the heights of this region were designed to be 50, 80, 110, 150, 180, 270 μm. Before imaging, we degassed the chip and all the solution with a vacuum pump for 5 min to achieve stable hour-long imaging. The microfluidic chip was pre-filled with KRBB solution (125 mM NaCl, 5.9 mM KCl, 2.56 mM CaCl₂, 1.2 mM MgCl₂, 1 mM L-glutamine, 25 mM HEPES, 0.1% BSA, pH 7.4) containing 3 mM D-glucose before use. Then we injected the islet into the microfluidic chip using a 10 μL pipette from the inlet shown in Fig.1a.

**Isolation and culture of mouse pancreatic islets**. Pancreatic islets were isolated from mice as previously described[75]. Briefly, mice were euthanized, and freshly-prepared collagenase P solution (0.5 mg/ml) was injected into the pancreas via the common bile duct. The perfused pancreas was digested at 37 °C for 17 min, and the islets were handpicked under a stereoscopic microscope. After isolation (defined as day 0), islets then were transferred to a dish RPMI 1640 medium (11879020, Gibco) supplemented with 10% fetal bovine serum (10099141 C, Gibco), 5.5 mM D-glucose, 100 unit/ml penicillin and 100 mg/ml streptomycin for overnight culture (generally at 5 p.m. at day 0). They were maintained at 37 °C and 5% CO₂ in a culture incubator before the imaging experiments. We do imaging on day 1 and 2 (10 a.m.–10 p.m.). The time between islet isolation and the experiment typically were 17 h–53 h.

**Dissociation into single cells**. After overnight culturing, the isolated islets were washed with HBSS (14175095, Gibco). They were then digested with 0.25% trypsin-EDTA (25300-062, Gibco) for 3 min at 37 °C followed by briefly shaking. The digestion was stopped by the culture medium, and centrifuged at 94 g for 5 min. The cells were suspended by RPMI 1640 culture medium. The cell

suspension was plated on coverslips in the poly-L-lysine-coated Glass Bottom Dish (D35-14-1-N, Cellvis) or the microfluidic chip. The dishes or chips were then kept for 40 min in the culture incubator at 37 °C, 5% $CO_2$ to allow cells to adhere. Additional culture medium was then added, and the cells were cultured for 24 h before the imaging experiments.

**Immunofluorescence of single islet and cells.** Immunofluorescence experiment and analysis of *Glu-Cre+; GCaMP6f^f/+*, and *Ins2-RCaMP1.07* mice were performed as previously described[75,76]. Islets were fixed in 4% Paraformaldehyde Fix Solution for 1 h, and were blocked overnight in PBS, and 0.3% Triton-100X at room temperature. The samples were incubated overnight days at 4 °C in a guinea pig anti-insulin antibody (1:200, A0564, Dako), a rat anti-somatostatin antibody (1:200, ab30788, Abcam), and a mouse anti-glucagon antibody (1:200, G2654, Sigma) separately. The islets were then incubated for 2 h at 4 °C separately with Goat anti-Guinea pig immunoglobulin G (IgG) (H+L) secondary antibody (DyLight™ 488) (1:1000, SA5-10094, Invitrogen), Goat anti-Guinea pig IgG H&L (DyLight™ 550) (1:1000, SA5-10095, Invitrogen), Goat anti-Rat IgG H&L (Alexa Fluor 568) (1:1000, A-11077, Invitrogen), Goat anti-Rat IgG H&L (Alexa Fluor 350) (1:1000, A-21093, Invitrogen), Goat anti-Mouse IgG H&L (Alexa Fluor 488) (1:1000, A32766, Invitrogen), and Goat anti-Mouse IgG H&L Cy3 (1:1000, A10521, Molecular Probes).

**Fluorescence imaging of $Ca^{2+}$.** For samples isolated from C57BL/6 J mice, we loaded the samples with Cal-520 AM dye (2 μM, 37 °C, 40 min) for $Ca^{2+}$ imaging experiments. In our experiments, we used two confocal spinning-disc microscopes. For *Glu-Cre+; GCaMP6f^f/+; Ins2-RCaMP1.07* mouse islets, we used Dragonfly 200 series (Andor) with a Zyla4.2 sCMOS camera (Andor), and the Fusion software (version 2.3.0.36). All channels were collected with a 40x/0.85 NA Microscope Objective (Warranty Leica HCX PL APO). For 1% transmission of 561 nm illumination (100 mW) and 3% transmission of 488 nm illumination (150 mW), the exposure was set to 20 ms and 100 ms, respectively. A single band pass filter with a center wavelength 525 nm (band width 50 nm) was used to collect the GCaMP6f emission. And single band pass filter with a center wavelength 600 nm (band width 50 nm) was used to collect the RCaMP1.07 emission. For other islets, we used an inverted TiE microscope (Nikon) attached to an UltraVIEW VOX system spinning disc (PerkinElmer) with a C9100-13 EMCCD camera (Hamamatsu), which was controlled by the Volocity software (PerkinElmer, 6.6.1) and equipped with a 40×/1.25 NA objective (Nikon CFI APO). For 2% transmission of 561 nm illumination (50 mW) and 20% transmission of 488 nm illumination (40 mW), the exposure was set to 20 ms and 100 ms, respectively. The islet in the microfluidic chip was kept at 37 °C and 5% $CO_2$ on the microscope stage during imaging. The reagents were automatically pumped into the microfluidic chip with a flow rate of 400 μL/h by the TS-1B syringe pump (LongerPump), which was controlled by software written in MATLAB.

**α Cell number counting after $Ca^{2+}$ imaging.** *GCaMP6f+* mice were generated by multiply rounds crossbreeding of *EIIa-Cre+;GCaMP6f^f* mice until spontaneous germline recombination was detected. The missing of stop codon between flox sites was confirmed by PCR using template tail DNA extracted from *EIIa-Cre−;GCaMP6f^f* mice. Then a triple transgenic mouse line (*Glu-Cre+;tdTomato^f/+;GCaMP6f+*) were generated by crossbreeding of *GCaMP6f+* and *Glu-Cre+;tdTomato^f/f* mouse. Using two-photon microscopy, we identified individual α cells in islets and recorded their 10 G stimulated $Ca^{2+}$ signal (Fig. 6a and b). The *spots* function in Imaris9.7 was used for cell identification and counting (with the parameter of 10 um cell diameter).

**Similarity index.** The similarity index (SI) was defined to compare the spatial activation similarity between two rounds of 10 G stimulation (3 seconds sampling interval). Firstly, we used differences between two consecutive frames to form new stacks, which best highlighted the times and cells of significant increases in fluorescence intensities (Fig. 1d). The similarity index was used to define how similar two rounds of stimulations evoked responses from the same location. We selected a fixed region of interest of 150 pixels × 150 pixels (45 μm × 45 μm). We made montages that are indicated by the Target (*T* in *Round1*) and Paired (*P* in *Round2*) images for cross-correlations calculation (the size of the *T*, and *P* were 150*150*t). We also randomly selected 50 regions of 150 pixels × 150 pixels in the second round image and indicated them as Random (R in *Round2*) to calculate the random similarity indexes.

The image similarity was calculated by the pixel-based SSIM index[77]. The SSIM index of *T* and *P* was indicated by $SSIM(T,P)$, and the mean SSIM index between *T* and *R* was indicated by $\overline{SSIM(T,R)}$. Because the perfect match conferred a SSIM index of 1, we defined the similarity index between *T* and *P* as the normalized SSIM index:

$$SI(T,P) = \frac{SSIM(T,P) - \overline{SSIM(T,R)}}{1 - \overline{SSIM(T,R)}} \quad (4)$$

According to Eq. (4), a similar pair of *T* and *P* gives large $SI(T, P)$. When *T* and *P* were from the identical position from two rounds of 10 G stimulation, $SI(T, P)$ belonged to the "round1, 2 pair" category shown in Fig. 1e. Otherwise, $SI(T, P)$ belonged to the "random pair" category shown in Fig. 1e.

**Feature definition and UMAP classification.** To further quantify each α-β $Ca^{2+}$ oscillation cycle pair, we defined 13 features, including 7 αβ features, 3 α features, and 3 β features (Supplementary Fig. 3).

**αβ features.** We first identified peaks of $Ca^{2+}$ transients in either α or β cells. We denoted the time when the traces peaked as $t_{\alpha2}$, $t_{\beta2}$, the time when the traces increased from basal values to half-maximal of the peaks as $t_{\alpha1}$, $t_{\beta1}$, and the times when the traces fall from peaks to half-maximal of peaks as $t_{\alpha3}$, $t_{\beta3}$. We denoted the times when the subsequent cycle of β cell peaked as $t_{\beta2}'$, the times when the traces increased from basal values to half-maximal of the peaks as $t_{\beta1}'$, and the times when the traces fall from peaks to half-maximal of peaks as $t_{\beta3}'$. The oscillation period *T* was defined as the time difference between two β peaks. $T_{\beta\alpha\_peak}$ (or $T_{\beta\alpha\_50}$) was defined as the peak (or half-maximal) waiting time for the activation of α cells after β cell activation. The peak (or half-maximal) waiting time for the activation of β cells after α cell activation was defined as $T_{\alpha\beta\_peak}$ (or $T_{\alpha\beta\_50}$). The phase difference ($\Delta\theta$) between α and β cell $Ca^{2+}$ was defined as the ratio of $T_{\beta\alpha\_peak}$ and *T*. The last feature was the ratio between $\alpha_{FWHM}$ and $\beta_{FWHM}$.

**α Cell features and β cell features.** The durations $\tau_{\alpha\_up}$ and $\tau_{\beta\_up}$ were defined as the length of times for the $Ca^{2+}$ to stay beyond the half-maximal of the peak. We fitted the fluorescence decay of α (or β) $Ca^{2+}$ traces with an exponential function to obtain the time constant $\tau_{\alpha\_decay}$ (or $\tau_{\beta\_decay}$). The full widths at half-maximal of peak fluorescence intensity of α and β cells were defined as $\alpha_{FWHM}$ and $\beta_{FWHM}$.

**UMAP oscillation pairs classification.** We collected 658 α-β cycle pairs from 21 islets with all of their features assembled into a feature matrix. Using the UMAP (Uniform Manifold Approximation and Projection) algorithm[49], these α-β cycle pairs were clustered into two distinct groups – a cluster of fast cycle pairs and a cluster of slow cycle pairs (Fig. 4b).

The previously mentioned fast, slow, and mixed oscillation modes are composed of these α-β cycle pairs in different ways. That is to say, the fast oscillation mode is purely composed of the fast cycle pairs, the slow oscillation mode of slow cycle pairs, and the mixed oscillation mode of a combination of fast and slow cycle pairs (Fig. 5c, right column).

**Mathematical modeling.** To understand the origin of the various oscillation modes and the factors controlling them, we developed a mathematical model incorporating the paracrine interactions between α and β cells.

**α-β Phase oscillator model.** In our model, an islet was simplified into two coupled oscillators - an α cell and a β cell (Fig. 5a). Each cell was described by a phase variable θ, which was driven by an intrinsic term and a paracrine term. G represented glucagon concentration in the system. $f_s$ ($\theta_\beta$) represented insulin secretion and $f_s$ ($\theta_\alpha$) glucagon secretion. τ is the degradation/dilution constant of glucagon. The equations were (1)-(3), where $\omega_\alpha$, $\omega_\beta$, $K_{\beta\alpha}$, $K_{\alpha\beta}$ are constants. $\omega_\alpha$, $\omega_\beta$ belong to the intrinsic term and $K_{\beta\alpha}$, $K_{\alpha\beta}$ belong to the paracrine term.

**Intrinsic term.** The intrinsic term described the oscillating periods of isolated single cells. As previous studies have shown, single β cells oscillate for a period of ~5 minutes (Supplementary Fig. 5a, b). In the model, the period of β cells was assumed to be 360 s $\left(\omega_\beta = \frac{2\pi}{360}\right)$. In contrast, the isolated α cells oscillated faster than the β cells (Supplementary Fig. 5c, d). Hence, we assumed the period of the α cell to be 40 s ($\omega_\alpha = \frac{2\pi}{40}$) in the model.

**Paracrine term.** The paracrine term described how cells were coupled by releasing hormones. As previous studies have shown, β cells inhibit α cell function[34,53], and α cells promote β cell function[26,27,31,41,61]. Thus, our model consisted of three parts, the coupling coefficients $K_{\alpha\beta}$ and $K_{\beta\alpha}$ (negative), hormone secretion function $f_s$ ($\theta$), and receptor binding response functions $f_{r\alpha}$ ($\theta$) and $f_{r\beta}$ ($\theta$).

**Hormone secretion function $f_s$ ($\theta$).** In the model, we assumed that cells released hormones periodically with phase changes—cells released hormones when they approached phase π, and stopped when they approached phase 0. As $f_s$ ($\theta$) represented the amount of hormone released by a single cell, it reached the minimum 0 at phase 0, and the maximum 1 at phase π. Between 0 and π, $f_s$ ($\theta$) monotonically increased; Between π and 2π, $f_s$ ($\theta$) monotonically decreased. In numerical simulations, we chose the trigonometric function $f_s(\theta) = \frac{1-\cos(\theta)}{2}$. We also discuss other functional forms of $f_s$ ($\theta$) in the robustness discussion section (Supplementary Figs. 6d and 7).

**α Cell response function $f_{r\alpha}$ ($\theta$).** β cells inhibit α cells through a variety of mechanisms, including directly by releasing insulin, $Zn^{2+}$, and GABA[34], or by indirectly activating δ cells to secrete somatostatin[53]. Despite diversified mechanisms, activation of β cells constantly hyper-polarized α cells and inhibited glucagon release[34].

In the model, $-f_{r\alpha}(\theta)$ described the inhibitory paracrine response of α cells (the negative sign here came from the definition of $K_{\beta\alpha}$ as a negative number). The function $-f_{r\alpha}(\theta)$ was negative in the interval of phase 0 to π, and became positive in the interval of phase π to 2π. When the α cell was inhibited, $-f_{r\alpha}(\theta)$ would push the α cell phase to around 0 or 2π, which would stably stay around phase 0 and stop glucagon secretion until the inhibitory hormone was relieved. In numerical simulation, we chose the trigonometric function $f_{r\alpha}(\theta) = \sin(\theta)$. We also discuss other functional forms of $f_{r\alpha}(\theta)$ in the robustness discussion section (Supplementary Figs. 6c and 7).

**β Cell response function $f_{r\beta}(\theta)$.** α cells promote β cells through the release of glucagon. Recent studies have shown that glucagon increases the level of cAMP in β cells by binding to GCGR and GLP-1 receptors[27,31,33,78]. In the model, $f_{r\beta}(\theta)$ described the stimulatory paracrine response of β cells. The function $f_{r\beta}(\theta)$ was positive in any phase. As long as glucagon was present, β cell $Ca^{2+}$ would pace at a faster rate. In numerical simulations, we chose the function $f_{r\beta}(\theta) = 1$. In the robustness discussion section, we will discuss other functional forms of $f_{r\beta}(\theta)$ (Supplementary Fig. 6b).

**Coupling coefficient $K_{\alpha\beta}$ and $K_{\beta\alpha}$.** $K_{\alpha\beta}$ and $K_{\beta\alpha}$ were critical parameters that described the magnitudes of the interaction between the α and β cells. We normalized the absolute values of $f_s(\theta)$, $f_{r\alpha}(\theta)$ and $f_{r\beta}(\theta)$ to between 0 and 1. The normalized functions described a unit amount of hormones released by a single cell expressing a unit amount of receptors. Therefore, $K_{\alpha\beta}$ is proportional to the product of the number of α cells and the number of receptors expressed on β cells. $K_{\beta\alpha}$ is proportional to the product of the number of β cells, the amount of hormone released by one β cell, and the number of receptors expressed on α cells. In Fig. 5d, $K_{\alpha\beta}$ was chosen to be between 0 and $\frac{2\pi}{10}$. This $\frac{2\pi}{10}$ value represents the assumption that the α cell can speed up the β cell into a period of 10 s with glucagon concentration of 1. $K_{\beta\alpha}$ was chosen to be between 0 and $-\frac{2\pi}{10}$. When $K_{\beta\alpha}$ is less than $-\frac{2\pi}{40}$, $\frac{d\theta_\alpha}{dt}$ can reach 0. Therefore, the β cell is able to arrest the α cell activity.

**Hormone concentration.** The stimulation from α cell to β cell depends on the glucagon concentration $(G)$ in the system. The glucagon concentration was dynamically determined by α cell secretion and the passive dilution/degradation. The inhibition from β cell to α cell was assumed to depending on the secreted insulin at the moment. Thus, the insulin concentration was proportional to β cell secretion.

**Specific model selected for numerical simulation.** In summary, we used the parameters and functions below for the numerical simulation in Fig. 5d. The intrinsic oscillation frequency of α cells was $\omega_\alpha = \frac{2\pi}{40}$, the intrinsic oscillation frequency of β cells was $\omega_\beta = \frac{2\pi}{360}$, the hormone release function was $f_s(\theta) = \frac{1-\cos(\theta)}{2}$, the α cell receptor function was $f_{r\alpha}(\theta) = \sin\theta$, and the β cell receptor function was $f_{r\beta}(\theta) = 1$. τ was set to 0.5 for the simulation presented in the main text. Therefore, the model is

$$\frac{d\theta_\alpha}{dt} = \omega_\alpha + K_{\beta\alpha}\frac{(1-\cos(\theta_\beta))}{2}\sin(\theta_\alpha) \tag{5}$$

$$\frac{d\theta_\beta}{dt} = \omega_\beta + K_{\alpha\beta}G \tag{6}$$

$$\frac{dG}{dt} = \frac{(1-\cos(\theta_\alpha))}{2} - \tau G. \tag{7}$$

**Phase-locking analysis.** To better understand the dynamics of our oscillator model, we introduce the concept of winding number in this section.

The winding number $w(K_{\alpha\beta}, K_{\beta\alpha})$ is geometrically equivalent to the slope of the trajectory in the α and β phase diagrams (Fig. 5b). It reflected the relative growth rates of the phase of α and β cells. For example, when α and β cells are 1:1 phase-locked, w = 1; when β cells strongly inhibit α cells, they are phase-locked by 0:1, $w = 0$; when α and β cells exhibited no interaction, $w = \frac{\omega_\beta}{\omega_\alpha}$. When β cells oscillate 2 mπ and α cells pass 2nπ, it belongs to a phase locking category with winding number $w = \frac{m}{n}$.

**Slow, fast and mixed regime.** In our model, the controllable parameters were $K_{\alpha\beta}$ and $K_{\beta\alpha}$. According to the phase locking index $w(K_{\alpha\beta}, K_{\beta\alpha})$, the paracrine coupling parameter space was separated into categories of weak coupled regime (phase locking index w > 1), slow regime (phase locking index w = 1/1), fast regime (w = 0/1), as well as mixed regime (0 < w < 1) (Fig. 5d).

**Robustness analysis.** There were five critical parameters and functions in the model, the intrinsic oscillation frequency $\omega_\alpha$ of the α cell, the intrinsic oscillation frequency $\omega_\beta$ of the β cell, hormone release function $f_s(\theta)$, α cell response function $f_{r\alpha}(\theta)$, and β cell response function $f_{r\beta}(\theta)$. To test the robustness of phase-locked results against different choices of functions, we changed the function forms and re-ran the numerical simulation. Similarly, we also changed the winding number $w(K_{\alpha\beta}, K_{\beta\alpha})$ (Supplementary Figs. 6 and 7).

**Hormone secretion function $f_s(\theta)$.** $f_s(\theta)$: The secretion function was changed to $f_s(\theta) = (\frac{1-\cos(\theta)}{2})^3$. The secretion function was always non-negative, with a value of 0 at phase 0 and a value of 1 at phase π. They increased monotonically from phase 0 to π and decreased monotonically from π to 2π. With updated function, the oscillation mode regimes still existed and displayed no significant difference (Supplementary Fig. 6d). The only difference was that the dynamics needed to be observed is in the narrowed parameter space.

**α Cell response function $f_{r\alpha}(\theta)$.** $f_{r\alpha}(\theta)$: The response function of α cells was changed to $f_{r\alpha}(\theta) = -sin^3(\theta)$, which is negative in the range of 0 to π and positive in the range of π to 2π. In the newly calculated model, all three oscillation modes still existed and showed no differences (Supplementary Fig. 6c).

**β Cell response function $f_{r\beta}(\theta)$.** $f_{r\beta}(\theta)$: The response function of β cells was changed to $f_{r\beta}(\theta) = abs(sin(\theta))$, which is always positive. In the newly calculated model, all three oscillation modes still existed and showed no differences (Supplementary Fig. 6b). The only difference was that the dynamics needed to be observed is in the enlarged parameter space.

**Elevated basal glucagon level in islet.** We considered the islet may have varied basal glucagon concentration (note the parameter b). And the newly secreted glucagon cyclically pushed β cells faster. In the model (8)-(10), an elevated basal glucagon concentration was equivalent to increasing the β-cell intrinsic oscillation frequency $(\omega_\beta)$. Supplementary Fig. 6e showed that these oscillators were still stably phase-locked when the β cell intrinsic period was increased to 3 min. All three oscillation modes still existed and showed no differences.

$$\frac{d\theta_\alpha}{dt} = \omega_\alpha + \frac{K_{\beta\alpha}(1-\cos(\theta_\beta))}{2}\sin(\theta_\alpha) \tag{8}$$

$$\frac{d\theta_\beta}{dt} = \omega_\beta + K_{\alpha\beta}(G + b) \tag{9}$$

$$\frac{dG}{dt} = \frac{(1-\cos(\theta_\alpha))}{2} - \tau G \tag{10}$$

**Hormone concentration approximation.** In the original model, the stimulation from α cell to β cell was mainly determined by the accumulation level of the glucagon $(G)$. The inhibition from β cell to α cell was instantaneous, depending on the secreted insulin at the moment $(f_s(\theta_\beta))$. When the interaction between the α cell and the β cell was instantaneous, the glucagon concentration was $f_s(\theta_\alpha)$, and the glucagon concentration was $f_s(\theta_\alpha)$. The model was

$$\frac{d\theta_\alpha}{dt} = \omega_\alpha + K_{\beta\alpha}\frac{1-\cos(\theta_\beta)}{2}\sin(\theta_\alpha) \tag{11}$$

$$\frac{d\theta_\beta}{dt} = \omega_\beta + K_{\alpha\beta}\frac{1-\cos(\theta_\alpha)}{2} \tag{12}$$

The result is shown in Supplementary Fig. 6g. Note it was similar to the original model, and in particular, the phase-locking phenomenon was preserved. When the concentrations of insulin and glucagon were determined by the accumulated level, the model became

$$\frac{d\theta_\alpha}{dt} = \omega_\alpha + K_{\beta\alpha}I\sin(\theta_\alpha) \tag{13}$$

$$\frac{d\theta_\beta}{dt} = \omega_\beta + K_{\alpha\beta}G \tag{14}$$

$$\frac{dG}{dt} = \frac{1-\cos(\theta_\alpha)}{2} - \tau G \tag{15}$$

$$\frac{dI}{dt} = \frac{1-\cos(\theta_\beta)}{2} - \tau I. \tag{16}$$

Here $I$ represented the insulin concentration. The result is shown in Supplementary Fig. 6h. Note these oscillators were still stably phase-locked. All oscillation modes still existed and showed no differences.

**Shift phase.** It was manually set that cells would secrete at phase π. Further analysis showed that this assumption would not alter the output of this system. Suppose $f_1$, $f_2$ were two periodic mono-increasing function maps from $[0,2\pi]$ to

[0,2$\pi$], and each controlled the $\beta$ and $\alpha$ cells. Then the model is

$$\frac{d\theta_\alpha}{dt} = \omega_\alpha + K_{\beta\alpha}\frac{1-\cos(f_1(\theta_\beta))}{2}\sin(f_2(\theta_\alpha)) \qquad (17)$$

$$\frac{d\theta_\beta}{dt} = \omega_\beta + K_{\alpha\beta}G \qquad (18)$$

$$\frac{dG}{dt} = \frac{1-\cos(f_2(\theta_\alpha))}{2} - \tau G. \qquad (19)$$

Denote their inverse function as $f_1^{-1}, f_2^{-1}$ respectively, e.g. $f_1^{-1}(\theta'_\beta) = \theta_\beta, f_2^{-1}(\theta'_\alpha) = \theta_\alpha$ .Then,

$$\frac{d\theta_\alpha}{dt} = \frac{df_2^{-1}(\theta'_\alpha)}{dt} = \frac{df_2^{-1}(\theta'_\alpha)}{d\theta'_\alpha}\frac{d\theta'_\alpha}{dt} \qquad (20)$$

$$\frac{d\theta_\beta}{dt} = \frac{df_1^{-1}(\theta'_\beta)}{dt} = \frac{df_1^{-1}(\theta'_\beta)}{d\theta'_\beta}\frac{d\theta'_\beta}{dt}. \qquad (21)$$

Substitute these into dynamics, we get

$$\frac{d\theta'_\alpha}{dt} = \frac{d\theta'_\alpha}{df_2^{-1}(\theta'_\alpha)}\left(\omega_\alpha + K_{\beta\alpha}\frac{1-\cos(\theta'_\beta)}{2}\sin(\theta'_\alpha)\right) \qquad (22)$$

$$\frac{d\theta'_\beta}{dt} = \frac{d\theta'_\beta}{df_1^{-1}(\theta'_\beta)}(\omega_\beta + K_{\alpha\beta}G) \qquad (23)$$

$$\frac{dG}{dt} = \frac{1-\cos(\theta'_\alpha)}{2} - \tau G \qquad (24)$$

If $f_i$ were a linear transformation, then $g'_i(\theta'_i)$ is constant. Specifically, phase shift meant $g'_i(\theta'_i) = 1$, so it would not alter the dynamic behavior of this model. If symmetry of secretion functions and response functions was slightly broken, e.g.

$$f_i(\theta) = \frac{\pi}{\theta^*}mod(\theta, 2\pi)\ \theta \in [0, \theta^*]\ i = 1, 2 \qquad (25)$$

$$f_i(\theta) = \frac{\pi}{2\pi - \theta^*}(mod(\theta, 2\pi) - 2\theta^* + 2\pi)\ \theta \in (\theta^*, 2\pi]\ i = 1, 2 \qquad (26)$$

It can be found that all three oscillation mode regimes still exist in the newly calculated models. Only the border was slightly deformed (Supplementary Fig. 7).

**$\alpha$ Cells decay time constants in mixed mode oscillation**. The experiment data showed that $\alpha$ cells had different decay time constants in the mixed-mode oscillation (Fig. 7g and h). Meanwhile, in the model, the mixed-mode oscillations were composed of the fast and slow $2\pi$ cycles. This section will discuss the relationship between model and experimental results.

In the fast mode oscillation case ($w = 0/1$), when $\theta_\beta$ completed a $2\pi$ cycle, $\theta_\alpha$ failed to activate completely and was repressed to phase 0 very quickly due to full activation of $\theta_\beta$. Therefore, the decrease of $\alpha$ cell $Ca^{2+}$ signal was determined by the inhibitory paracrine effect. In experiments, the $\alpha$ cells displayed a faster decay time constant.

In the slow mode oscillation case ($w = 1/1$), $\theta_\alpha, \theta_\beta$ alternately completed a $2\pi$ cycle and $\theta_\beta$ took much more time to repress $\theta_\alpha$. Therefore, the decrease of $\alpha$ cell $Ca^{2+}$ signals was mainly determined by its own internal clearance rate. Experimentally the $\alpha$ cells had a slower decay time constant.

For the mixed mode oscillation case ($0 < w = m/n < 1$), $\theta_\alpha$ completed fewer cycles than $\theta_\beta$. When $\theta_\alpha$ ran less than a $2\pi$ cycle while $\theta_\beta$ completed a $2\pi$ cycle ($f^{k+1}(\theta_\alpha) - f^k(\theta_\alpha) < 2\pi$), the decay rate was fast. Otherwise, the decay rate was slow. Therefore, we experimentally observed both fast and slow decay time constants. In fact, the slow and the fast decay constants display clear patterns in their ordering – fast decays followed by a slow decay.

**Statistics and reproducibility**. Statistical comparisons in Fig. 7c and e were conducted using a two-tailed paired $t$-test (Graphpad PRISM 7.0). Statistical comparisons in Fig. 2g, h, and Table 1 were conducted using a two-tailed non-paired t-test (Graphpad PRISM 7.0). See the two-tailed P values in the Source Data file. Symbols based on the $p$ values were defined as n.s. $p > 0.1$, *$p < 0.05$, **$p < 0.01$, ***$p < 0.001$, ****$p < 0.0001$. Experiments shown in Figs. 2b, 3e, f have repeated five times independently with similar results; Fig. 6b for three times; Fig. 1d and Supplementary Fig. 1a and b for five times; Supplementary Fig. 1c and d for three times and Supplementary Fig. 2c for three times.

**Reporting summary**. Further information on research design is available in the Nature Research Reporting Summary linked to this article.

## Data availability
The authors declare that all data supporting the findings of this study are available from the corresponding author on reasonable request. Source data in Figs. 1b, c, e, 2c–h, 3a–d, f, 4b–d, 6a, b, 7c–f and h are provided as a Source Data file. Source data are provided with this paper. A reporting summary for this Article file is available as a Supplementary Information file. Source data are provided with this paper.

## Code availability
ImageJ1.52e was used to process the fluorescent images and extract the mean fluorescent intensity. MATLAB (2019a) build-in function "findpeaks" is used to extract the features for each oscillation. "ode23" function is used to simulate the mathematical equations. The custom-made MATLAB GUI script that supports the finding of phase-locked $\alpha$ and $\beta$ cells in this study are available in GitHub at https://github.com/nevaehRen/Islet.

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

## Acknowledgements

We are grateful to Erik Gylfe for many discussions and critical reading of the manuscript. We thank Anders Tengholm, Yanmei Liu, and Louis Tao for helpful discussions. We thank Chunxiong Luo and Shujing Wang for their help with the microfluidic chip design. We thank Daniel Tang and Iain Bruce for manuscript editing. We thank Xiaowei Chen and Xiao Yu for sharing mice lines. We are grateful to the Imaging Core Facility and the State Key Laboratory of Membrane Biology for assistance with two-photon microscopy. We thank Dr. Ye Liang for providing the technical support of Imaris software. This work was supported by grants from the National Natural Science Foundation of China (12090053, 32088101, 81925022, 92054301), The National Key Research and Development Program of China (2018YFA0900700, 2021YFF1200500), NSFC Innovation group projects (31821091) and the Beijing Natural Science Foundation (Z20J00059, 7152079, 5194026). H.R. was supported by the Boya Postdoctoral Fellowship of Peking University. X.P. was supported by the Postdoctoral Fellowship of Peking-Tsinghua Center for Life Science.

## Author contributions

C.T. and L.C. conceived and supervised the study, and wrote the manuscript. H.R. led the project, designed experiments, carried out data analysis and mathematical modeling, and wrote the manuscript. Y.L. designed and manufactured the microfluidic chip, designed and performed experiments, and wrote the manuscript. C.H. designed and performed experiments, and wrote the manuscript. X.Y. contributed to Figure preparation and experiment supervision, Y.Y and K.S. contributed to the mathematical modeling, B.S., T.Z., S.W. and X.P. contributed to the experiments and data analysis.

## Competing interests

The authors declare no competing interests.
