## [Peer review file · Nature Communications]

REVIEWER COMMENTS

Reviewer #1 (Remarks to the Author):

General:

This paper proposes and tests by both experiments and modeling the hypothesis that the frequency of in vitro calcium oscillations in the islets of Langerhans is regulated by the strength of paracrine coupling between the beta and alpha cells in the islet. These cells secrete insulin and glucagon respectively, which both play critical roles in the pathogenesis and treatment of diabetes, both type 1 and type 2. The oscillations are known to be important for optimal action of insulin on its target tissues, especially the liver, but also muscle and adipose tissue. The role of the wide range of frequencies, covering an order of magnitude, from tens of seconds to several minutes, in contrast, is not well understood, but it is ubiquitous across species, suggesting that it is probably important. It is also an important puzzle that has thus far eluded a full understanding despite a mature field of mathematical modeling devoted to it. It is also a paradigmatic phenomenon for the general field of biological oscillations. The author's hypothesis stands in contrast to the more typical view that frequency is controlled by intrinsic properties of the beta cells, so this paper would be an important contribution.

Specific Comments:

1. One point the authors could emphasize more is the surprising finding that glucagon and insulin both oscillate at high glucose. This has been observed before (<https://pubmed.ncbi.nlm.nih.gov/25911612/>), but it is at odds with the classic picture that glucagon raises glucose when it is low while insulin lowers it when it is high. More recent work by the Campbell and Wess groups has shown that glucagon assists insulin in lowering high glucose by enhancing insulin secretion, and this study sheds further light on how this happens. See also point #2.
2. l. 120, "two types of alpha cells": there may not be two discrete types but rather a broad distribution of alpha-cell properties that results in two classes of response, inhibition by glucose and activation by glucose.
3. l. 161: The mean period of the slow oscillations was 104 s, which is less than half that seen by others, such as the Henquin, Tengholm, and Satin groups. This should be acknowledged.
4. l. 268, "adding glucagon did not affect fast oscillating islets": This is at odds with classic observations by Cook and Ikeuchi (<https://pubmed.ncbi.nlm.nih.gov/6087071/>) who found up to a threefold increase in frequency in fast islets. Please cite this.
5. l. 320: "secretin" is not the right word, perhaps "secreted factors"
6. l. 354 and elsewhere: a detailed calculation is shown in Fig. 5E of m:n phase locking (Arnold tongues). Examples are shown with $n > 1$ (the mixed pattern), but what about $m > 1$?
7. l. 356: Would the authors care to speculate about what benefit phase-locking to different frequencies might confer to islets? Or is this just an epiphenomenon?
8. l. 365, "both intrinsic properties of single cells and interactions among different cell types may contribute": The authors should mention here or elsewhere some of the known intrinsic mechanisms that have strong effects on frequency, such as TEA, which at low doses mainly blocks BK K(Ca) channels; acetylcholine, which increases frequency by increasing ER calcium efflux; and, most recently discovered, pyruvate kinase activation (see <https://pubmed.ncbi.nlm.nih.gov/33147484/>).

Reviewer #2 (Remarks to the Author):

In this manuscript, Ren et al characterize Ca²⁺ oscillation patterns in alpha and beta cells in mouse islets and present a model attempting to explain why some islets predominantly show fast Ca²⁺ oscillations whereas others show slow oscillations in response to glucose stimulation. The authors demonstrate that the response pattern is inherent to a particular islet and remains constant during repeated glucose challenges. Using mice with transgenic expression of different Ca²⁺ reporters in alpha and beta cells, it is demonstrated that Ca²⁺ oscillations triggered by high glucose concentrations in the two cell types are synchronized and phase shifted, and that the pattern can be modulated by glucagon. A mathematical model is presented that takes intercellular coupling strength via paracrine mechanisms into account and which reproduces the main features of the experimental data.

The study addresses a long-standing, yet unresolved, question about what determines the pattern of Ca²⁺ oscillations in mouse islets. Clarifying the determinants of the oscillation patterns is relevant for understanding the regulation of insulin and glucagon secretion and should be of interest to a broad readership as altered islet hormone secretion patterns are hallmarks of diabetes.

The authors have taken an elegant methodological approach allowing impressive simultaneous recordings of Ca²⁺ in alpha and beta cells within intact islets. Some of the results mirror previously published data and the main novelty is the finding that there is strict phase-locking of Ca²⁺ oscillations with alpha cells showing a constant lag of ~20 s in relation to beta cells, most likely caused by paracrine suppression of alpha cell activity. The detailed quantitative analyses overall constitute a strength but there is a tendency that the data is overanalyzed, which makes the manuscript less accessible without necessarily providing important new insights.

Specific:

One of the key messages is that beta cells show a "tunable delay" after alpha cells (T-alpha-beta), whereas the delay between beta and alpha cells (T-beta-alpha) remains constant. The constant T-beta-alpha is – just as the authors describe – consistent with an inhibitory factor that suppresses alpha cells. The factor could emanate from both beta and delta cells and the authors discuss the potential role of delta cells, although the model only includes one inhibitory cell type. However, the "tunable" T-alpha-beta is more difficult to understand. The parameter scales with the period of oscillations (T), i.e. it is longer in the case of a slow oscillating islet and shorter in fast islets. I am not convinced by the authors' conclusion that the value of the phase shift determines T as if there were a causal relationship. It seems that it might equally well be the other way around. If T is determined by other factors, and T-beta-alpha is constant, it follows that T-alpha-beta must change with T.

The authors show that the T-alpha-beta parameter takes many minutes to stabilize. Also constant glucagon supply favors fast oscillations, and that is the case even in individual cells. It is therefore not clear that there would be a requirement for feedback from alpha to beta cells within the time frame of one oscillation cycle. The value of the T-alpha-beta parameter and its role to determine the oscillation mode therefore remains unclear.

Having said that, I do not question the conclusion that glucagon promotes fast oscillations. It is well-established and has been observed in different laboratories. However, there is no evidence that this would be the only factor that determines whether the islets will respond with fast or slow oscillations. There is an extensive literature with mathematical models that describes the fast and slow patterns by beta-cell intrinsic variables, for example from the Sherman, Bertram and Satin team.

If paracrine input from alpha cells were the only determinant of the oscillation mode, it may appear surprising that only 50% of the fast-oscillating islets responded to the combination of insulin and

glucagon/GLP-1 receptor antagonists with a switch to slow oscillations. This relatively weak effect is discussed on lines 329-334 but I do not understand the link the authors make to the glucagon receptor knockout data and the insulin data.

The authors describe two types of alpha cells mirroring previous reports. The majority of cells are shown to be initially inhibited when glucose is increased and a small fraction of cells are stimulated. Judged from the traces in Fig 2F, these effects are not very convincing, mainly because the alpha cell activity at low glucose is low in the brief period shown. In the trace exemplifying inhibited alpha cells, the most striking effect is the pronounced Ca²⁺ spiking after a few minutes when the beta cells begin to oscillate.

Fig S5, panel D: What does the alpha cell response pattern look like in the presence of 3 mM glucose between the high glucose challenges?

Line 118: The results description gives the impression that cells are stimulated with high glucose in the presence of 25 mM KCl but that is not evident from the figure. Please clarify.

The data from db/db mice do not add anything to support the conclusions of the study. The results are used to claim that there is a critical role of Ca²⁺ oscillations for normal hormone release and glucose homeostasis but such a conclusion would require more detailed characterization of this particular strain of db/db mice.

What is the time between islet isolation and the experiments? It is known that this may affect the oscillation mode in mouse islets.

Line 40: I would rephrase the statement that "dampening and disappearance of islet Ca²⁺ oscillation is an early biomarker in the pathogenesis of type 2 diabetes". Most of the studies the authors cite discuss hormone pulsatility and there is limited evidence for changes in Ca²⁺ oscillations. Depending on how "biomarker" is defined, it may be discussed whether cytosolic Ca²⁺ oscillations can qualify. Circulating concentrations of hormones can for sure.

The numbers of islets are given but please also specify from how many mice and independent isolations.

There are a number of language issues and small mistakes:

Line 47: "Only slow Ca²⁺ oscillations are mostly seen..." (contradictory)

Line 111: "Under resting glucose stimulation..." (specify glucose concentration instead)

Lines 112-113: "...the mean Ca²⁺ transients were flat..." (unclear what is meant)

Line 114: "...elevated glucose stimulation..." (specify glucose concentration)

Line 116: "...alpha responder..." (alpha cells?)

Line 124: "different pools of alpha cells" (why pools and not just different alpha cells?)

Lines 132-133: "tightly inhibited alpha cells originating from their neighboring beta cells" (difficult to understand)

Fig S1, panel B, middle: Text "2.2% alpha cells" should likely read "2.2% beta cells".

Fig S1, panel C and D: How many cells were analyzed?

Fig S5, panel B: It seems that Ca²⁺ is increasing before the increase of glucose. Probably the rectangles have been shifted.

Fig S7: Panel labels A, B and C are missing.

Reviewer #3 (Remarks to the Author):

COMMENTS TO AUTHOR:

This manuscript reported a new method to reproduce the heterogeneous Ca^{2+} oscillation patterns in islet micro-tissue by using microfluidic chip device and transgenic mice. The presented method had proved that the phase shift between α and β cells set the islet oscillation period and pattern. It appears the results seem interesting, however, several uncertainties in the data need to be clarified and major revision is necessary before it can be justified for publishing in NC. The specific comments are listed below:

1. In this work, the authors introduced four models of Ca^{2+} oscillation in islet as shown in figure 1B. however, only one oscillation mode was displayed in figure 1D. how about other three kinds of Ca^{2+} oscillation mode in the islet. they should be provided as well.
2. In the part of introduction, the authors should properly cover the literature about the study of Ca^{2+} oscillation patterns in pancreatic islet micro-tissue in animal, human, or any other models at present.
3. It is difficult to see the illustration of the microfluidic chip in figure 1A. what is the working principle? how to manipulate the islet in the device. More detailed information are missing.
4. The authors identified the labeled pancreatic α - and β -cells in transgenic mice by immunofluorescence staining. The image quality of insulin and GCG staining to identify the cell types are poor (figure S1). The data could not be used for qualifying the specific cell types.
5. On page 4 (paragraph 2), the author claimed "the existence of phase-locked pattern was found in islet cells". why? It is not clear why α -cells are stably and globally phase-locked to β cells under elevated glucose stimulation.
6. In figure 3, the additional data is required to describe the phase-locked pattern of islet in diabetes mice. It is essential to analyze the difference of islet function in normal and pathological conditions. especially, what type of communications are coordinated among α - and β -cells in islet cell types under normal and disease states.
7. The authors established the mathematical modeling to study the communications between different cell types in islet, how these modes could be utilized to track and evaluate the diabetes?
8. There are several spelling errors through the text, such as in reference 2. please check it carefully.

Point-by-point response to reviewers' comments

Index Table |. The list of modified figures.

Revised index	New/Modified
Figure	
Fig. 1B	Modified
Figs. 2E and 2F	Modified
Figs. 4C, 4D and 4E	Modified
Figs. 5B, 5D, 5E and 5F	Modified
Fig. 6A	New
Figs. 6D, 6F and 6H	Modified
Supplementary figure	
Figs. S1A, S1B	New
Fig. S1E	Modified
Fig. S4	Modified
Fig. S5C and S5D	Modified
Fig. S6	Modified
Fig. S7	Modified
Fig. S8	Modified
Fig. S9F	Modified

REVIEWER COMMENTS

Reviewer #1 (Remarks to the Author):

General:

This paper proposes and tests by both experiments and modeling the hypothesis that the frequency of in vitro calcium oscillations in the islets of Langerhans is regulated by the strength of paracrine coupling between the beta and alpha cells in the islet. These cells secrete insulin and glucagon respectively, which both play critical roles in the pathogenesis and treatment of diabetes, both type 1 and type 2. The oscillations are known to be important for optimal action of insulin on its target tissues, especially the liver, but also muscle and adipose tissue. The role of the wide range of frequencies, covering an order of magnitude, from tens of seconds to several minutes, in contrast, is not well understood, but it is ubiquitous across species, suggesting that is probably important. It is also an important puzzle that has thus far eluded a full understanding despite a mature field of mathematical modeling devoted to it. It is also a paradigmatic phenomenon for the general field of biological oscillations. The author's hypothesis stands in contrast to the more typical view that frequency is controlled by intrinsic properties of the beta cells, so this paper would be an important contribution.

Response: We appreciate the reviewer's positive and encouraging comments.

Specific Comments:

1. One point the authors could emphasize more is the surprising finding that glucagon and insulin both oscillate at high glucose. This has been observed before (<https://pubmed.ncbi.nlm.nih.gov/25911612/>), but it is at odds with the classic picture that glucagon raises glucose when it is low while insulin lowers it when it is high. More recent work by the Campbell and Wess groups has shown that glucagon assists insulin in lowering high glucose by enhancing insulin secretion, and this study sheds further light on how this happens. See also point #2.

Response 1: Thank you for raising this point and for the suggested reference. We have also found that glucagon potentiates insulin secretion via GCGR and GLP-1R (Zhang, Yulin, et al. 2021). We have added these references and the corresponding discussion in the paper (**Page 9, line 327**).

38
39
40
41
42
43
44
45
46
47
48
49
50

2. I. 120, "two types of alpha cells": there may not be two discrete types but rather a broad distribution of alpha-cell properties that results in two classes of response, inhibition by glucose and activation by glucose.

Response 2: Sorry for the confusion. We have redefined those α cells as "excited α cell" that demonstrated a pronounced and slow Ca^{2+} transient which was delayed compared to that in the β cells before showing phase-locked Ca^{2+} oscillations with β cells, and those as "inhibited α cell" cells that didn't show the initial peak. We have plotted the histogram for the maximal amplitude of Ca^{2+} transients of α cells within the first pronounced glucose-stimulated Ca^{2+} transient in β -cells (**Fig.R1**). There were indeed two populations of responses (84% and 16%), centered at ~6% and ~300% maximal Ca^{2+} elevation, respectively. We are investigating the underlying mechanism as a future project. We have revised the manuscript accordingly (**Page 4, line 114**).

51
52
53
54
55
56
57
58
59
60
61
62
63
64
65
66
67
68
69
70

Fig. R1. We classified two types of α cells based on their initial Ca^{2+} responses to high glucose stimulation. In the left panel, we showed normalized mean glucose-stimulated Ca^{2+} transients from β cells as the reference (the red trace, dash line indicated 50% activation of β -cell, grey box shows the duration within which α cell Ca^{2+} peak is measured). We also showed two types of responses from α cells: the classical "inhibited α cell" that demonstrated little responses during this period (light green), and an "excited α cell" that showed a large response (dark green). The right panel showed the distribution of the maximal amplitude of α cell Ca^{2+} signal within this period ($n = 267$ α cells from 5 islets from 3 mice).

3. I. 161: The mean period of the slow oscillations was 104 s, which is less than half that seen by others, such as the Henquin, Tengholm, and Satin groups. This should be acknowledged.

Response 3: Thank you for the reminder. Per your suggestion, we have summarized periods of slow islet oscillations from the mentioned groups in Table R1 for comparison. In our experiments, we observed the slow oscillations with a mean period of 104s (**Fig.1B**, **Fig.3F**, and **Fig.4D**, ranged 1-3 min), which Henquin and Satin groups have also seen. However, we failed to see slow oscillations with mean periods at the 5-10 min range, as Satin and Tengholm groups have shown previously. We reasoned that this might be associated with the incubation of islets in microfluidic chips and the perfusion rate (400 ul/hour). We have added a discussion and the reference in the paper (**Page 9, line 349**).

Oscillation period	Glucose concentration	Reference	
~2 min	10G	J.C. Henquin et al., 1995	
2-4 min and 5-8 min	11G	L.S. Satin et al., 2009	
~2 min	11G	L.S. Satin et al., 2016	Figure 5D
1-10 min, 5 in 9 islets within 3 min	11G	L.S. Satin et al., 2019	Figure 2C
3-7 min and 3 min	11G	Anders Tengholm et al., 2009	Table 1 and Figure 2A
7 min	20G	Anders Tengholm et al., 2015	

71 **Table R1. Typical Oscillation Periods from Previous Islet Study.**

72
73
74
75
76
77
78
79
80
81

4. I. 268, "adding glucagon did not affect fast oscillating islets": This is at odds with classic observations by Cook and Ikeuchi (<https://pubmed.ncbi.nlm.nih.gov/6087071/>) who found up to a threefold increase in frequency in fast islets. Please cite this.

Response 4: We used 100 nM glucagon in our experiments, much lower than Cook's study (2 μ M), and that did not affect fast oscillating islets. After you have pointed out this discrepancy, we did try to co-apply 2 μ M glucagon with 10 mM glucose to the islet. However, we still failed to see consistent accelerations of Ca^{2+} oscillation in fast islets (**Fig.R2**). We have added a discussion and the reference in the revised manuscript (**Page 9, line 319**).

82
83
84
85
86
87
88
89
90
91
92
93
94
95
96
97
98
99
100
101
102
103
104
105
106

Fig. R2. No obvious acceleration in oscillation frequency is observed by glucagon application in fast islets. Left panel: Mean islet Ca^{2+} signal with 10G stimulation in the absence (black) and presence (blue) of glucagon. Right panel: Oscillation period with 10G stimulation before, during, and after glucagon application.

5. I. 320: "secretin" is not the right word, perhaps "secreted factors"

Response 5: Thank you. We have changed the word on **Page 8, Line 314**.

6. I. 354 and elsewhere: a detailed calculation is shown in Fig. 5E of m:n phase locking (Arnold tongues). Examples are shown with $n > 1$ (the mixed pattern), but what about $m > 1$?

Response 6: We have shown the 2:1 phase-locking example ($m=2, n=1$) as case 4 in **Fig. 5D** (bottom row).

7. I. 356: Would the authors care to speculate about what benefit phase-locking to different frequencies might confer to islets? Or is this just an epiphenomenon?

Response 7: We have speculated that stable phase-locking to different frequencies and modes could ensure a stable and tunable ratio of insulin and glucagon secretion (**Page 9, Line 341**). Currently, we are combining islet Ca^{2+} imaging with real-time detection of α and β cell secretion to investigate the physiological roles of the phase-locking. However, since we don't have robust data yet, we did not elaborate on the topic further in the current paper.

8. I. 365, "both intrinsic properties of single cells and interactions among different cell types may contribute": The authors should mention here or elsewhere some of the known intrinsic mechanisms that have strong effects on frequency, such as TEA, which at low doses mainly blocks BK K(Ca) channels; acetylcholine, which increases

107 frequency by increasing ER calcium efflux; and, most recently discovered, pyruvate kinase activation
108 (see <https://pubmed.ncbi.nlm.nih.gov/33147484/>).

109
110 **Response 8:** Thank you. We have added such discussions on **Page 10, Line 351**. It reads now " We note that the
111 oscillation period of β cells can be influenced by many factors, such as BK K(Ca) channel blockage, enhanced ER
112 calcium efflux, and the recently discovered pyruvate kinase activation"

113
114
115
116 Reviewer #2 (Remarks to the Author):
117

118 In this manuscript, Ren et al characterize Ca²⁺ oscillation patterns in alpha and beta cells in mouse islets and
119 present a model attempting to explain why some islets predominantly show fast Ca²⁺ oscillations whereas others
120 show slow oscillations in response to glucose stimulation. The authors demonstrate that the response pattern is
121 inherent to a particular islet and remains constant during repeated glucose challenges. Using mice with transgenic
122 expression of different Ca²⁺ reporters in alpha and beta cells, it is demonstrated that Ca²⁺ oscillations triggered by
123 high glucose concentrations in the two cell types are synchronized and phase shifted, and that the pattern can be
124 modulated by glucagon. A mathematical model is presented that takes intercellular coupling strength via paracrine
125 mechanisms into account and which reproduces the main features of the experimental data.

126
127 The study addresses a long-standing, yet unresolved, question about what determines the pattern of Ca²⁺
128 oscillations in mouse islets. Clarifying the determinants of the oscillation patterns is relevant for understanding the
129 regulation of insulin and glucagon secretion and should be of interest to a broad readership as altered islet hormone
130 secretion patterns are hallmarks of diabetes.

131
132 The authors have taken an elegant methodological approach allowing impressive simultaneous recordings of Ca²⁺ in
133 alpha and beta cells within intact islets. Some of the results mirror previously published data and the main novelty is
134 the finding that there is strict phase-locking of Ca²⁺ oscillations with alpha cells showing a constant lag of ~20 s in
135 relation to beta cells, most likely caused by paracrine suppression of alpha cell activity. The detailed quantitative
136 analyses overall constitute a strength but there is a tendency that the data is overanalyzed, which makes the
137 manuscript less accessible without necessarily providing important new insights.

138
139 **Response 9:** Thank you for your appreciation of our work and the comment about the tendency of overanalyzing
140 data. We have seriously taken your advice in making the revisions.

141
142
143 Specific:
144

145 One of the key messages is that beta cells show a "tunable delay" after alpha cells (T-alpha-beta), whereas the delay
146 between beta and alpha cells (T-beta-alpha) remains constant. The constant T-beta-alpha is – just as the authors
147 describe - consistent with an inhibitory factor that suppresses alpha cells. The factor could emanate from both beta
148 and delta cells and the authors discuss the potential role of delta cells, although the model only includes one
149 inhibitory cell type. However, the "tunable" T-alpha-beta is more difficult to understand. The parameter scales with the
150 period of oscillations (T), i.e. it is longer in the case of a slow oscillating islet and shorter in fast islets. I am not
151 convinced by the authors' conclusion that the value of the phase shift determines T as if there were a causal
152 relationship. It seems that it might equally well be the other way around. If T is determined by other factors, and T-
153 beta-alpha is constant, it follows that T-alpha-beta must change with T. The authors show that the T-alpha-beta
154 parameter takes many minutes to stabilize. Also constant glucagon supply favors fast oscillations, and that is the
155 case even in individual cells. It is therefore not clear that there would be a requirement for feedback from alpha to
156 beta cells within the time frame of one oscillation cycle. The value of the T-alpha-beta parameter and its role to
157 determine the oscillation mode therefore remains unclear.

158
159 **Response 10:** Thank you for these critical insights. We now have revised our mathematical phase-locking model in
160 the main text. In the new model, the stimulation from α cell to β cell depends on the overall level of the glucagon in
161 the system, which is determined by the accumulation due to the continuous by pulsatile secretion and the
162 degradation/dilution of glucagon (**Fig. 5B**). In contrast, the stable $T_{\beta\alpha}$ suggests that the inhibition from β cell to α cell
163 is strong and instantaneous. In the supplemental materials, we have analyzed both cases in which the interaction
164 between the α cell and the β cell was determined by the instantaneous secretion of the paracrine factors as in our
165 previous model (**Fig. S6G, Methods, Page 8, Line 297**), and in which the paracrine interaction is determined by the
166 accumulated levels of these factors (**Fig. S6H, Page 8, Line 304**). The results were similar, and in particular, the
167 phase-locking phenomenon was quite robust. We have added the model part in the main text (**Page 5, Line 190 and**
168 **Page 6, Line 202**) and in the Methods section (**Page 4, line 159 and Page 6, line 225**).

169
170 Having said that, I do not question the conclusion that glucagon promotes fast oscillations. It is well-established and
171 has been observed in different laboratories. However, there is no evidence that this would be the only factor that
172 determines whether the islets will respond with fast or slow oscillations. There is an extensive literature with
173 mathematical models that describes the fast and slow patterns by beta-cell intrinsic variables, for example from the
174 Sherman, Bertram and Satin team. If paracrine input from alpha cells were the only determinant of the oscillation
175 mode, it may appear surprising that only 50% of the fast-oscillating islets responded to the combination of insulin and
176 glucagon/GLP-1 receptor antagonists with a switch to slow oscillations. This relatively weak effect is discussed on
177 lines 329-334 but I do not understand the link the authors make to the glucagon receptor knockout data and the
178 insulin data.

179
180 **Response 11:** Thank you. As others have studied, intrinsic variables may affect the fast and slow oscillation patterns.
181 The partial suppression of fast-oscillating islets in responding to the combination of insulin and glucagon/GLP-1
182 receptor antagonists was also puzzling to us. However, we have shown that using a combination of glucagon/GLP-1
183 receptor antagonists failed to suppress islet insulin release to the same extent as glucagon KO and antagonizing
184 GLP-1 receptor (Zhang, Yulin, et al. 2021). This suggests that we may not have efficiently abolished paracrine effects
185 within the closely-packed islets. We have added a discussion about this in the discussion section (**Page 9, Line 319**).

186
187 On the other hand, based on the following experiments, we argue that stimulative effects other than the
188 paracrine input from α cells played only a minor role in our experimental conditions. In our hands, we seldom
189 observed fast Ca^{2+} oscillations in isolated single β cells (**Fig.S5A-B**). Moreover, we have generated a triple
190 transgenic mouse line (Glu-Cre^+ ; $\text{tdTomato}^{f/+}$; GCaMP6f^+). Using two-photon microscopy, we can identify individual α
191 cells in islets and record the 10G stimulated Ca^{2+} signals. We found that the Ca^{2+} activities of islets at 10G were
192 highly correlated with their α cell numbers (**Fig.R3**). Interestingly, islets with ≤ 30 α cells all showed slow oscillations;
193 on the other hand, most islets with more than 30 α cells showed typical fast oscillations. Therefore, these data
194 strongly support the model, and show that paracrine contribution from islet α cells is associated with rapid Ca^{2+}
195 oscillations in islet β cells (Given the Glu-cre labeling efficiency is around $\sim 50\%$, the actual threshold would be around
196 ~ 60 α cells). We have added this result in the main text (**Page 7, Line 254**), **Figs. 6A** and **6B**, and in Methods (**Page**
197 **2, Line 89**).

198

Fig. R3. α Cell Abundant Islets Showed Fast Oscillations and α Cell Deficient Islets Showed Slow Oscillations.

A. Scatter plot of islet's α cell number and mean islet oscillation period under 10G stimulation ($n = 30$ islets from 4 mice). The dots with yellow edge were data from those islets shown in **C**. The islets were isolated from *Glu-Cre⁺; tdTomato^{+/+}; GCaMP6f⁺* mice. **B.** α cell number (left) and islet diameter (right) for the slow and fast oscillating islet groups. **C.** 3D images of six islets selected from **A** (imaged with two-photon microscopy). First and third columns: Red showed the tdTomato expressing α cells, and green showed the GCaMP6f expressing whole islet cells. Second and fourth columns: Mean Ca^{2+} signal with 40 min 10G stimulation. The number of α cells (tdTomato positive cells) and the diameter of the islet are shown in each graph.

The authors describe two types of alpha cells mirroring previous reports. The majority of cells are shown to be initially inhibited when glucose is increased and a small fraction of cells are stimulated. Judged from the traces in Fig 2F, these effects are not very convincing, mainly because the alpha cell activity at low glucose is low in the brief period shown. In the trace exemplifying inhibited alpha cells, the most striking effect is the pronounced Ca^{2+} spiking after a few minutes when the beta cells begin to oscillate.

Response 12: Thank you for the reminder, we have changed **Fig. 2F** to show the entire 5 min trace of these islet cells in 3G. As shown in the new **Fig. 2F** and **Fig. R4**, islet α cells were either active or inactive at 3G, and were all inactive for a short period after the switching of high glucose and before the first phase β cell activation (see **Fig. 2E**).

We have redefined those α cells as "excited α cell" that demonstrated a pronounced and slow Ca^{2+} transient which was delayed compared to that in the β cells before showing phase-locked Ca^{2+} oscillations with β cells, and those as "inhibited α cell" cells that didn't show the initial peak. We have plotted the histogram for the maximal amplitude of Ca^{2+} transients of α cells within the first pronounced glucose-stimulated Ca^{2+} transient in β -cells (**Fig. R1**). There were indeed two populations of responses (84% and 16%), centered at $\sim 6\%$ and $\sim 300\%$ maximal Ca^{2+} elevation, respectively. We are investigating the underlying mechanism as a future project. We have revised the manuscript accordingly (**Page 4, line 114**).

226
227

228
229
230
231
232
233
234
235
236
237
238
239
240
241
242

Fig. R4. β and α Cell Ca^{2+} signal under 3G and 10G stimulation. Two β cell Ca^{2+} signals (red) and eight α cell Ca^{2+} signals (green).

Fig S5, panel D: What does the alpha cell response pattern look like in the presence of 3 mM glucose between the high glucose challenges?

Response 13: Fig. R5 showed the α cell Ca^{2+} signal under 3G and 10G stimulation. We have added the 3G-10G-3G-10G response of α cell in Fig. S5D. Isolated single α cells exposed to 3 mM glucose exhibited distinct responses: half of them (53 of 104 cells) were active under 3G conditions, among which most (47 of 53 cells) were also stimulated by 10 mM glucose (Fig. R5A, Left). In contrast, there were 25% α cells (29 of 104 cells) that only responded to 10 mM glucose (Fig. R5A, Right). While most of the responses of α cells to 3G-10G stimulation were consistent, some cells (14 of 80) demonstrated different patterns to the same stimulus delivered a second time (Fig. 5RC). This result has been updated in Fig. S5D.

243
 244
 245
 246
 247
 248
 249
 250
 251
 252
 253
 254
 255
 256

Fig. R5. α Cell Ca^{2+} signal under 3G and 10G stimulation. **A.** α cells under 3G (15 min) and 10G (30 min) stimulation. 1st column: 3G active and 10G active α cells (47 of 104 cells). 2nd column: 3G inactive and 10G active α cells (29 of 104 cells). **B.** α cell under repetitive 3G (15 min) and 10G (45 min) stimulations. 1st row: 3G active and 10G active α cells in both rounds of stimulation (29 of 80 cells). 2nd row: 3G inactive and 10G active α cells in both rounds of stimulation (14 of 80 cells). 3rd row: α cells responded differently in two rounds of stimulation (14 of 80 cells).

Line 118: The results description gives the impression that cells are stimulated with high glucose in the presence of 25 mM KCl but that is not evident from the figure. Please clarify.

Response 14: Sorry for the confusion. KCl was added in the end to benchmark responses from all responsive α cells. **Fig. 2H** showed the 3G and 10G active ratio (normalized with the number of KCl responsive cells). We have changed our expression on **Page 4, Line 120**.

Fig. R6. Islet α Cell Ca^{2+} Signal Under 3G and 10G Stimulation. First row: 3G and 10G inactive α cell (n = 83 in 198 cells). Second row: 3G active and 10G inactive α cell (n = 9 in 198 cells). Third row: 3G inactive and 10G active α cell (n = 47 in 198 cells). Forth row: 3G and 10G active α cell (n = 59 in 198 cells). Islets were isolated from *Glu-Cre⁺;GCaMP6f^{+/+}* mice (n = 198 α cells from 12 islet from 4 mice).

The data from db/db mice do not add anything to support the conclusions of the study. The results are used to claim that there is a critical role of Ca^{2+} oscillations for normal hormone release and glucose homeostasis but such a conclusion would require more detailed characterization of this particular strain of db/db mice.

Response 15: Thank you. Per your suggestion, we have removed the db/db mice data and left the disease model study for future papers.

What is the time between islet isolation and the experiments? It is known that this may affect the oscillation mode in mouse islets.

Response 16: After isolation (defined as day 0), islets were transferred to a dish containing 5.5G culture media for overnight incubation (generally at 5 p.m. on day 0). We do imaging on day 1 and day 2 (10 a.m. – 10 p.m.). The time between islet isolation and the experiment typically was 17 h – 53 h. Thank you for the reminder. We have added this information to the Methods section on **Page 2, Line 46**.

Line 40: I would rephrase the statement that "dampening and disappearance of islet Ca^{2+} oscillation is an early biomarker in the pathogenesis of type 2 diabetes". Most of the studies the authors cite discuss hormone pulsatility and there is limited evidence for changes in Ca^{2+} oscillations. Depending on how "biomarker" is defined, it may be discussed whether cytosolic Ca^{2+} oscillations can qualify. Circulating concentrations of hormones can for sure.

Response 17: We have changed our expression to "The dampening and disappearance of islet Ca^{2+} oscillation are associated with the pathogenesis of diabetes" on **Page 2, line 40**. Thank you.

The numbers of islets are given but please also specify from how many mice and independent isolations.

Response 18: The number of mice has been added in the manuscript (**Page 3 line 83, Page 4 line 115, Page 5 line 171, Page 11 line 405, Page 12 line 432 / 435, Page 14 line 512, Page 15 line 533 / 533**) and the **Supplemental figure legends (Page 2 line 47, page 3 line 72 and page 4 line 102 / 104 / 115 / 117 / 119)**. Limited by the number of transgenic mice, we killed only one mouse with the same gene type (except C57 mice) in every isolation. The independent isolation number is equal to the number of mice.

293
294
295
296
297
298
299
300
301
302
303
304
305
306
307
308
309
310
311
312
313
314
315
316
317
318
319
320
321
322
323
324
325
326
327
328
329
330
331
332
333
334
335
336
337
338
339
340
341
342
343
344
345
346
347
348
349
350
351
352
353
354

There are a number of language issues and small mistakes:
Line 47: "Only slow Ca²⁺ oscillations are mostly seen..." (contradictory)

Response 19: Thank you very much for your careful reading. We have changed the expression on **Page 2, Line 48**. It reads now, "Slow Ca²⁺ oscillations are mostly seen in isolated β cells".

Line 111: "Under resting glucose stimulation..." (specify glucose concentration instead)

Response 20: We have changed the expression on **Page 3, Line 110**. It reads now, "Under 3 mM glucose stimulation ..."

Lines 112-113: "...the mean Ca²⁺ transients were flat.." (unclear what is meant)

Response 21: We have changed the expression on **Page 3, Line 112**. It reads now, "the mean Ca²⁺ intensity of β cells were low ..."

Line 114: "...elevated glucose stimulation..." (specify glucose concentration)

Response 22: We have changed the expression on **Page 3, Line 113**. It reads now, "Upon 10 mM glucose stimulation ..."

Line 116: "...alpha responder..." (alpha cells?)

Response 23: We have changed the expression on **Page 4, Line 114**. It reads now "... α cells ..."

Line 124: "different pools of alpha cells" (why pools and not just different alpha cells?)

Response 24: We have changed the expression on **Page 4, Line 125**. It reads now "... all glucose-responsive islet α cells became synchronized ..."

Lines 132-133: "tightly inhibited alpha cells originating from their neighboring beta cells" (difficult to understand)

Response 25: We have changed the expression on **Page 4, Line 131**. It reads, "Consistently we observed global α cell activation after the turning-off of β cells."

Fig S1, panel B, middle: Text "2.2% alpha cells" should likely read "2.2% beta cells".

Response 26: We have changed it in **Fig. S1B**.

Fig S1, panel C and D: How many cells were analyzed?

Response 27: 102 RCaMP1.07+ cells from 47 *Ins2-RCaMP1.07* islets and 303 tdTomato+ cells from 88 *Glu-Cre⁺;tdTomato^{f/+}* islets are analyzed for insulin immunofluorescence staining. 174 RCaMP1.07+ cells from 51 *Ins2-RCaMP1.07* islets and 119 tdTomato+ cells from 22 *Glu-Cre⁺;tdTomato^{f/+}* islets are analyzed for glucagon immunofluorescence staining. The number of islets, genetically labeled cells and IF overlapped has been added in **Fig. S1E-F** and the supplemental figure legends (**Page 1, lines 20 and 23**).

Fig S5, panel B: It seems that Ca²⁺ is increasing before the increase of glucose. Probably the rectangles have been shifted.

Response 28: The protocol was 20 min 3G + 40 min 10G + 20 min 3G + 40 min 10G. We have changed it in **Fig. S5B**.

Fig S7: Panel labels A, B and C are missing.

Response 29: We have changed it in **Fig. S7**.

Reviewer #3 (Remarks to the Author):

355
356
357
358
359
360
361
362
363
364
365
366
367
368
369
370
371
372
373
374
375
376
377
378
379
380
381
382
383
384
385
386
387
388
389
390
391
392
393
394
395
396
397
398
399
400
401
402
403
404
405
406
407
408
409
410
411
412

COMMENTS TO AUTHOR:

This manuscript reported a new method to reproduce the heterogeneous Ca²⁺ oscillation patterns in islet micro-tissue by using microfluidic chip device and transgenic mice. The presented method had proved that the phase shift between α and β cells set the islet oscillation period and pattern. It appears the results seem interesting, however, several uncertainties in the data need to be clarified and major revision is necessary before it can be justified for publishing in NC. The specific comments are listed below:

1. In this work, the authors introduced four models of ca²⁺ oscillation in islet as shown in figure 1B. however, only one oscillation mode was displayed in figure 1D. how about other three kinds of ca²⁺ oscillation mode in the islet. they should be provided as well.

Response 30: Thank you. We have added the mixed and slow oscillation modes in **Fig. S1A and B**.

2. In the part of introduction, the authors should properly cover the literature about the study of Ca²⁺ oscillation patterns in pancreatic islet micro-tissue in animal, human, or any other models at present.

Response 31: Per your suggestion, we have updated the introduction (**Page 2, line 39**). It reads now, "The oscillatory activity of pancreatic islets is widespread in many species, such as mouse and human. The dampening and disappearance of islet Ca²⁺ oscillation are associated with the pathogenesis of diabetes."

3. It is difficult to see the illustration of the microfluidic chip in figure 1A. what is the working principle? how to manipulate the islet in the device. More detailed information are missing.

Response 32: Thank you, we have added more details in the Methods section (**Page 1, line 32**). It reads, "Microfluidic chips were fabricated using the elastomer polydimethylsiloxane (PDMS). Briefly, we used the photo-polymerizable epoxy resin (SU-8-2100) to make a positive relief master, and the PDMS mold was cured on the master. PDMS mold was removed from the master as the channeled substrate. Then we used an oxygen plasma treatment to bond the PDMS mold with a glass coverslip (24 x 60 mm), as shown in detail in **Fig. 1A**. The islet trapping region was designed as a stair-like channel, using six different thicknesses of SU-8 photoresist. To trap islets of different sizes, the heights of this region were designed to be 50, 80, 110, 150, 180, 270 μ m. Before imaging, we degassed the chip and all the solution with a vacuum pump for 5 min to achieve stable hour-long imaging. The microfluidic chip was pre-filled with KRBB solution (125 mM NaCl, 5.9 mM KCl, 2.56 mM CaCl₂, 1.2 mM MgCl₂, 1 mM L- glutamine, 25 mM HEPES, 0.1% BSA, pH 7.4) containing 3 mM D-glucose before use. Then, we injected the islet into the microfluidic chip using a 10 μ L pipette from the inlet shown in **Fig.1A**."

4. The authors identified the labeled pancreatic α - and β -cells in transgenic mice by immunofluorescence staining. The image quality of insulin and GCG staining to identify the cell types are poor (figure S1). The data could not be used for qualifying the specific cell types.

Response 33: We repeated the islet immunofluorescence experiment and updated the results in **Fig. S1**.

5. On page 4 (paragraph 2), the author claimed "the existence of phase-locked pattern was found in islet cells". why? It is not clear why α -cells are stably and globally phase-locked to β cells under elevated glucose stimulation.

Response 34: Sorry for the confusion; we have removed this part from the manuscript.

6. In figure 3, the additional data is required to describe the phase-locked pattern of islet in diabetes mice. It is essential to analyze the difference of islet function in normal and pathological conditions. especially, what type of communications are coordinated among α - and β - cells in islet cell types under normal and disease states.

Response 35: Thank you for the advice, which is indeed our long-term goal. However, different diabetic models seem to have similar manifestations of disturbed Ca²⁺ signals. For example, in *ins-Cre^{+/+};Gcamp^{+/+}* transgenic mice on the HFD for two months, 10 mM glucose stimulation evoked a large Ca²⁺ transient followed by a long-lasting platform in islet β cells (**Fig. R8**). On the other hand, HFD islets exhibited regular oscillations under 7G stimulation, suggesting that HFD diet may shift the threshold of islet oscillating activity to lower glucose concentrations. With db/db mice data,

413 these indicate that changes in islet functions and glucose-stimulated Ca^{2+} oscillations in diabetes are complex and
414 need further characterization. Therefore, per your and reviewer 2's suggestion, we removed the disease data from
415 the current manuscript, leaving it to our future study.
416

417 **Fig. R8. HFD Feeding Increased the Glucose Threshold Sensitivity of Islet Oscillatory Activity.**
418 Mean Ca^{2+} signal of islets under 10G (first column) and 7G stimulation (second column). The islets were isolated
419 from 2 m HFD fed *ins-Cre^{+/+};Gcamp^{f/+}* mice.
420
421

422 7. The authors established the mathematical modeling to study the communications between different cell types in
423 islet, how these modes could be utilized to track and evaluate the diabetes?
424

425 **Response 36:** In addition to our responses above, recent studies have also shown that the islets from diabetes mice
426 would lose coordinated oscillations (<https://pubmed.ncbi.nlm.nih.gov/26943366/>). Therefore, it is necessary to extend
427 our simplified two-oscillator model into a multiple-oscillator interacting model to explain how the loss of coordination
428 affects the behavior of α and β cells. Therefore, it may be appropriate to leave the diabetes characterizations and
429 modeling to the forthcoming papers.
430

431 8. There are several spelling errors through the text, such as in reference 2. Please check it carefully.
432

433 **Response 37:** Revised. Thank you. See also **Responses 18-29** to Reviewer #2 above.
434

REVIEWERS' COMMENTS

Reviewer #1 (Remarks to the Author):

I am satisfied with the changes in response to the first round of review. The new and revised figure panels have clarified the results, and the new histogram (Fig. 6A) showing the strong dependence of islet frequency on the number of alpha cells in the islet is valuable new data.

I noticed on mistake: reference #62 has the right title but wrong authors, journal and page numbers. It looks like two different references were mixed together.

Reviewer #2 (Remarks to the Author):

The authors have responded well to most of the reviewer comments and made a thorough revision of the manuscript. Amendment of some minor, remaining, issues would further improve the text.

1. Lines 41-42. As pointed out in my first review, there is not much information about Ca^{2+} oscillations and diabetes and the selected references (with one exception) discuss hormone pulsatility. I suggest rephrasing the sentence to "Dampening and disappearance of hormone pulsatility and islet Ca^{2+} oscillations are associated with the pathogenesis of diabetes".

2. Line 120: The description of the KCl response is still unclear. It reads as if many cells do not respond to KCl but I have the feeling that the authors rather mean that among all cells that respond to KCl, many are inactive at both low and high glucose concentrations.

3. Line 202: Remove parenthesis around this paragraph.

4. The term " Ca^{2+} oscillations" in plural is more appropriate when describing the signaling phenomenon in general than singular " Ca^{2+} oscillation". Check e.g. lines 38, 41, 49, 74, 174 etc.

5. Methods, line 300. The word "I" appears less appropriate than "we" since all authors must take responsibility for all parts of the manuscript.

1 Point-by-point response to reviewers' comments

2 3 REVIEWER COMMENTS

4
5 Reviewer #1 (Remarks to the Author):

6
7 I am satisfied with the changes in response to the first round of review. The new and revised
8 figure panels have clarified the results, and the new histogram (Fig. 6A) showing the strong
9 dependence of islet frequency on the number of alpha cells in the islet is valuable new data.

10
11 I noticed on mistake: reference #62 has the right title but wrong authors, journal and page
12 numbers. It looks like two different references were mixed together.

13
14 **Response:** Thank you for the remind, reference #62 has been corrected.

15
16 Reviewer #2 (Remarks to the Author):

17
18 The authors have responded well to most of the reviewer comments and made a thorough
19 revision of the manuscript. Amendment of some minor, remaining, issues would further improve
20 the text.

21
22 1. Lines 41-42. As pointed out in my first review, there is not much information about Ca²⁺
23 oscillations and diabetes and the selected references (with one exception) discuss hormone
24 pulsatility. I suggest rephrasing the sentence to "Dampening and disappearance of hormone
25 pulsatility and islet Ca²⁺ oscillations are associated with the pathogenesis of diabetes".

26
27 **Response:** Line 39 has been corrected as " Dampening and disappearance of hormone
28 pulsatility, and islet Ca²⁺ oscillations are associated with the pathogenesis of diabetes."

29
30 2. Line 120: The description of the KCl response is still unclear. It reads as if many cells do not
31 respond to KCl but I have the feeling that the authors rather mean that among all cells that
32 respond to KCl, many are inactive at both low and high glucose concentrations.

33
34 **Response:** Line 120 has been corrected as "In addition, some α cells remained silent
35 both at the 3G and 10G conditions. All the α cells responded to 25 mM KCl."

36
37 3. Line 202: Remove parenthesis around this paragraph.

38
39 **Response:** The parenthesis in Line 202 has been removed.

40
41 4. The term "Ca²⁺ oscillations" in plural is more appropriate when describing the signaling
42 phenomenon in general than singular "Ca²⁺ oscillation". Check e.g. lines 38, 41, 49, 74, 174 etc.

43
44 **Response:** We have changed to "Ca²⁺ oscillations" in lines 38, 41, 49, 74, and 174.

45

46 5. Methods, line 300. The word "I" appears less appropriate than "we" since all authors must
47 take responsibility for all parts of the manuscript.

48

49 **Response:** The word "I" in line 676 is a symbol for the variable describing insulin
50 concentration. We have now changed it to italic format.

51